# Genome-scale pan-cancer interrogation of lncRNA dependencies using CasRx

Juan J. Montero [1,2,7] ✉, Riccardo Trozzo [1,2,7], Maya Sugden [1,2], Rupert Öllinger [1,2], Alexander Belka[1,2], Ekaterina Zhigalova [1,2], Paul Waetzig[1,2], Thomas Engleitner[1,2], Marc Schmidt-Supprian[2,3,4], Dieter Saur [2,3,5,6] & Roland Rad [1,2,3] ✉

Although long noncoding RNAs (lncRNAs) dominate the transcriptome, their functions are largely unexplored. The extensive overlap of lncRNAs with coding and regulatory sequences restricts their systematic interrogation by DNA-directed perturbation. Here we developed genome-scale lncRNA transcriptome screening using Cas13d/CasRx. We show that RNA targeting overcomes limitations inherent to other screening methods, thereby considerably expanding the explorable space of the lncRNAome. By evolving the screening system toward pan-cancer applicability, it supports molecular and phenotypic data integration to contextualize screening hits or infer lncRNA function. We thereby addressed challenges posed by the enormous transcriptome size and tissue specificity through a size-reduced multiplexed gRNA library termed Albarossa, targeting 24,171 lncRNA genes. Its rational design incorporates target prioritization based on expression, evolutionary conservation and tissue specificity, thereby reconciling high discovery power and pan-cancer representation with scalable experimental throughput. Applied across entities, the screening platform identified numerous context-specific and common essential lncRNAs. Our work sets the stage for systematic exploration of lncRNA biology in health and disease.

Deep characterization of the human transcriptome revealed that protein-coding sequence accounts for only a minority of the transcriptional output[1–5]. Among the most abundant RNA species are lncRNAs[6,7], which display highly cell-type-specific expression patterns[8,9]. The total number of annotated lncRNAs is roughly 4.5 times higher than that of protein-coding genes[6,10], yet only a few have been studied mechanistically so far. This highlights the need for approaches to systematically functionalize the lncRNAome.

Genome-scale perturbation of the protein-coding genome has been achieved through indel/frame-shift causation by DNA-targeting clustered regularly interspaced short palindromic repeats (CRISPR) systems[11–14]. However, these cannot be used to interrogate lncRNAs, as they lack an open reading frame. Creating large lncRNA deletions by CRISPR-associated protein 9 (Cas9) can address this problem[15] but suffers from limitations, such as low efficiency and extensive collateral perturbation of overlapping coding and regulatory sequence.

[1]Institute of Molecular Oncology and Functional Genomics, School of Medicine, Technische Universität München, Munich, Germany. [2]Center for Translational Cancer Research (TranslaTUM), School of Medicine, Technische Universität München, Munich, Germany. [3]German Cancer Consortium (DKTK), German Cancer Research Center (DKFZ), Heidelberg, Germany. [4]Institute of Experimental Hematology, School of Medicine, Technical University of Munich, Munich, Germany. [5]Department of Medicine II, Klinikum rechts der Isar, School of Medicine, Technische Universität München, Munich, Germany. [6]Institute for Experimental Cancer Therapy, School of Medicine, Technische Universität München, Munich, Germany. [7]These authors contributed equally: Juan J. Montero, Riccardo Trozzo. ✉e-mail: jj.montero@tum.de; roland.rad@tum.de

These issues can be partly addressed through Cas9-based targeting of lncRNA splice sites[16], which, however, is applicable only to multiexonic transcripts and requires availability of a protospacer adjacent motif (PAM) sequence close to splice sites. Finally, as discussed later in detail, double-strand break (DSB) causation by Cas9 and related 'toxicity'[17–25], is a confounding factor that is particularly detrimental for the analysis of lncRNA screens[26].

Another method used for lncRNA screening is CRISPR interference (CRISPRi)[27], which acts through the recruitment of transcriptional repressors to regulatory sequence[28]. For single guide RNA (sgRNA) design, identification of transcription start sites (TSSs) is imperative but often challenging due to incomplete/inaccurate lncRNA annotation[29]. Moreover, a main challenge to CRISPRi screening is created by repressive effects on neighboring regulatory sequences[27,30,31], making a substantial part of the lncRNAome inaccessible to CRISPRi-based interrogation.

These issues can be potentially overcome by direct RNA targeting. RNA interference (RNAi) is one option, but is limited by high off-target rates[32] and low lncRNA targeting efficacy. The latter is due to the nuclear localization of most lncRNAs, whereas the RNA-induced silencing complex exerts mainly cytoplasmic activity[33]. Antisense oligonucleotides, which function in the nucleus, cannot be encoded genetically as they are DNA-based.

An attractive solution to these drawbacks is offered by type VI CRISPR–Cas (Cas13) RNAases. Among them, CasRx displays the strongest efficiency[34]. CasRx has been used for small-scale perturbations[35,36], but genome-wide screening has not been achieved. Continuous RNA degradation requires permanent CasRx expression, which can be achieved by genome integration. However, the amount of protein produced is lower compared with that produced by episomal vectors. High CasRx amounts are needed to degrade the numerous RNA molecules. This differs from Cas9-based DNA cleavage, which relies on (permanent) editing of only two DNA molecules per cell. At the same time, too high CasRx activity can be detrimental to a screen, due to potentially increased risk of indiscriminate off-target RNA cleavage[37,38].

Another challenge is the incomplete annotation of the lncRNA transcriptome, which affects library design and target discovery[29]. In addition, the large number and tissue-specific expression of lncRNAs hamper the design of a cross-entity screening approach[1,8,9]. Indeed, Cas9–CRISPRi-based lncRNA screens used cell-line-specific libraries[15,16,27].

Here we developed a CasRX-based platform for genome-scale screening. Applied to the lncRNA transcriptome, we show that this approach overcomes limitations inherent to other modes of perturbation. We developed a screening strategy supporting high-throughput mapping of lncRNA dependencies across cancers. Its use identified known and new essential lncRNAs, including some that are common or cell-line specific.

## Results

### Optimization of a genome-integrated CasRx system

To overcome limitations of lncRNA targeting by Cas9 or CRISPRi, we developed a screening approach based on permanent expression of genome-integrated CasRx. We initially tested lentiviral systems for CasRx delivery but achieved only modest RNA targeting efficiencies, probably due to inadequate CasRx expression. We therefore cloned a 'CAG-NLS-CasRx-NLS-P2A-blasticidin' cassette flanked by Sleeping Beauty and PiggyBac inverted transposon repeats into an episomal vector (Fig. 1a), supporting plasmid-to-genome cassette mobilization by either transposase. To achieve multicopy genome integration, we cotransfected this plasmid with an expression vector for the most active PiggyBac transposase variant (hyPBase; Fig. 1a) at a 5:1 molar ratio (Fig. 1b).

For downstream screening purposes, we equipped human cancer cell lines from three tumor types with the CasRx system: glioblastoma

(LN-18, LN-119), nonsmall cell lung cancer (A549, NCI-H460) and pancreatic ductal adenocarcinoma (MIA PaCa-2, KP-4). We expanded single-cell clones from the original cell pools to identify clones with the highest CasRx expression and to minimize screening noise and artefacts linked to cellular heterogeneity[14]. CasRx clones are highly similar to their parental lines (Fig. 1c and Supplementary Fig. 1a) and are transposition-incompetent as nonautoreplicative episomal transposase vector gets lost during clonal expansion (Supplementary Fig. 1b). Quantitative insertion site sequencing (QiSeq)[39,40] detected 1–17 CasRx insertions per cell clone (Supplementary Fig. 1c).

To quantify CasRx activity in blasticidin-resistant cells, we expressed an unstable version of green fluorescent protein (GFP) and transduced cells with either GFP-targeting gRNA or control NT gRNA (Fig. 1d,e). GFP fluorescence (as measured by flow cytometry) decreased by 70–90% in cells receiving the GFP-targeting gRNA, confirming efficient RNA knockdown across these cell lines (Fig. 1f).

### RNA targeting without indiscriminate off-target cleavage

When first described in bacteria, Cas13 systems displayed off-target effects due to indiscriminate RNA degradation[41,42]. Such effects had not been observed in initial reports using CasRx in mammalian cells[34,43], but have been described recently in connection with very high episomal CasRx expression[37,38]. Indiscriminate off-target cleavage happens after on-target RNA degradation, during which Cas13 undergoes conformational changes, exposing its catalytic domain[44]. To interrogate whether our system suffers from such a problem, we performed three types of experiments.

First, we used fluorescent 'sensors,' as described previously[37]. We delivered GFP and tRFP657 to our cell lines and used CasRx to target GFP. Whereas the GFP signal decreased substantially, there no change in tRFP657 fluorescence was detectable in any of our cell lines, suggesting that our system does not display indiscriminate off-target cleavage (Fig. 1g). Second, we performed knockdown experiments targeting GFP or an endogenous protein-coding gene, followed by transcriptome sequencing. Differential expression analysis between targeting and NT conditions showed a lack of global gene dysregulation, which would be expected in scenarios of indiscriminate off-target cleavage (Fig. 1h,i). Third, we performed fitness screens using a control library including 1,634 NT gRNA pairs as well as gRNAs targeting 300 always-essential (AE) genes and 300 never-essential (NE) genes (1,200 gRNA pairs each). The NE genes display robust expression (Supplementary Fig. 1d) but no fitness phenotypes in corresponding CRISPR or RNAi screens[45] (Supplementary Fig. 1e). Since indiscriminate off-target cleavage is coupled to on-target cleavage, it would be expected in cells harboring NE-gRNAs, but not in cells carrying NT gRNAs. Importantly, we observed no differences in logarithmic fold change (LFC) distributions between NE and NT gRNAs (Fig. 1j). Altogether, these results provide evidence for a lack of indiscriminate collateral cleavage in our system.

### Target selection for lncRNA screens across solid tumors

We aimed to create a genome-scale sgRNA library for screening applications across solid tumors. Challenges toward this goal are: (1) the incomplete survey and annotation of the human lncRNA transcriptome (a critical hurdle for new discoveries and library design), (2) its large size (the number of lncRNA genes vastly exceeds the number of protein-coding genes, causing screening scalability issues) and (3) tissue-specific expression of lncRNAs (which complicates the design of a screening library for pan-cancer studies).

To address these challenges, we used human lncRNAs annotated in RNAcentral[7,46], the most comprehensive database for noncoding RNA sequences. We complemented this collection with evolutionarily conserved lncRNAs[47–52], resulting in 577,475 human transcripts (Fig. 2a). Transcripts and transcript isoforms in RNAcentral are not linked to individual lncRNA genes. To classify lncRNA species into lncRNA genes consisting of highly similar isoforms, we developed a computational

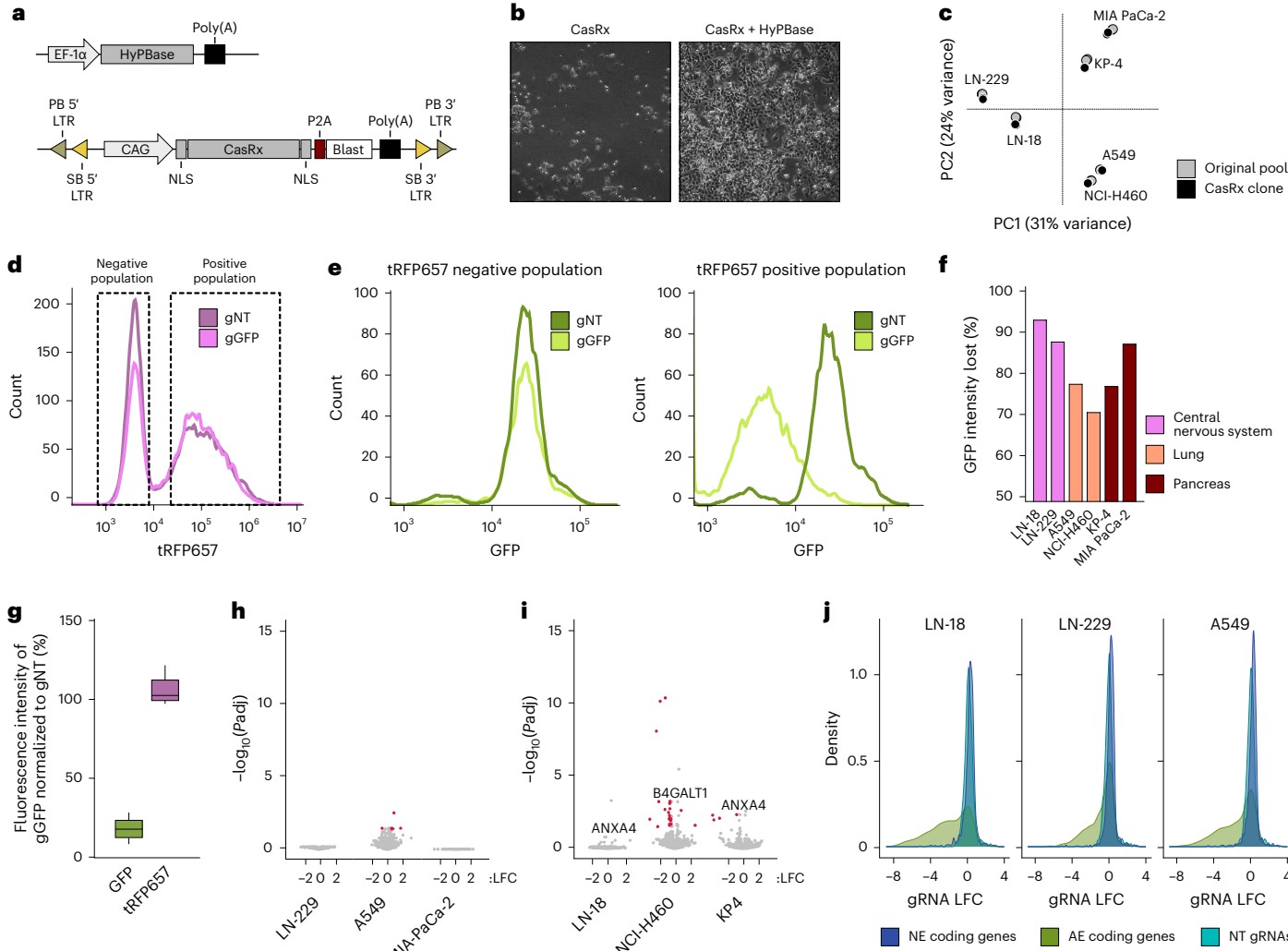

**Fig. 1 | Optimization of a genome-integrated CasRx system. a**, Schematic representation of the episomal HyPBase transposase and the transposon vectors used to deliver CasRx into mammalian cells. **b**, Representative brightfield microscope images of KP-4 cells 2 weeks after blasticidin selection. The puromycin control was performed once in each of the six generated cell lines. **c**, PCA plot of the 10% most variable genes using RNA-seq data from parental cell lines and derivative CasRx clones used in this study. **d**, tRFP657 intensity measured by flow cytometry in KP-4 cells, stably expressing GFP, transduced with a NT gRNA (gNT) or a GPF-targeting gRNA (gGFP). Dashed boxes indicate the extent of the negative and positive populations. **e**, GFP intensity measured by flow cytometry in the same cells as in **d**. Left panel, GFP intensity distribution of the tRFP657 negative cells; right panel, GFP intensity distribution of the tRFP657 positive cells. **f**, Percentage of CasRx activity in different cells. **g**, Boxplot showing the percentage of GFP or tRFP intensity between GFP-tRFP-positive cells transduced with a gGFP gRNA and cells transduced with a gNT gRNA.

The experiments were performed in all the CasRx cell lines used for screens; $n = 6$ independent cell lines. Boxplot data is presented as follows: center, median; box bounds, 25% and 75% percentile; whisker, 1.5× interquartile range (IQR). Outliers are marked as independent dots. **h**, Lack of global gene dysregulation upon on-target cleavage of stably expressed GFP. Volcano plots display whole-transcriptome data for indicated CasRx clones transduced with gGFP or gNT. **i**, Lack of global gene dysregulation upon on-target cleavage of endogenous transcripts. Volcano plots display whole-transcriptome data for indicated CasRx clones transduced with gRNAs targeting indicated protein-coding genes or gNT. $P$ values were computed with DESeq2 using the Wald test, and the FDR adjusted $P$ value ($P$adj) was calculated with the Benjamini–Hochberg method. **j**, LFC distribution of gRNAs for three screens performed using a CasRx control library. The library is composed of gRNAs targeting NE expressed coding genes (dark blue), NT gRNAs (light blue) and gRNAs targeting AE coding genes (green). Blast, blasticidin; LTR, long terminal repeats; PB, PiggyBac; SB, Sleeping Beauty.

pipeline that collapses transcript-based genomic occupancy, exonic overlap and transcript directionality (Supplementary Fig. 2a–c). These analyses gave rise to 97,817 lncRNA genes (Fig. 2a, b). Hence, our workflow allowed us to define and discriminate independent putative functional units, thereby providing a basis for rational design of sgRNA libraries (see Fig. 2c for an example).

We used the 97,817 lncRNA genes to designate targets for a screening library that optimally combines several critical criteria: (1) screening feasibility and scalability (to enable large-scale pan-cancer studies), (2) comprehensive representation of functional lncRNAs and (3) applicability across tumor entities. To define the library composition, we devised a four-step lncRNA selection process, which incorporates

considerations on potential functional relevance (indicated by the level of expression and/or cross-species conservation) and context dependencies (tissue-, cell- and genetic-context-dependent expression patterns). First, we selected 9,300 lncRNA genes with the highest average expression across the 839 solid tumor cell lines. Second, to avoid underrepresentation of specific tumor types, we selected additional 9,306 lncRNA genes with the highest expression in each tissue type. Third, in a similar approach, we next chose 8,892 lncRNA genes with the highest expression in each cell line, thereby accounting for cell line-specific or genetic context dependencies. Fourth, we enriched the above selection with 3,012 evolutionary conserved lncRNA genes (Fig. 2a).

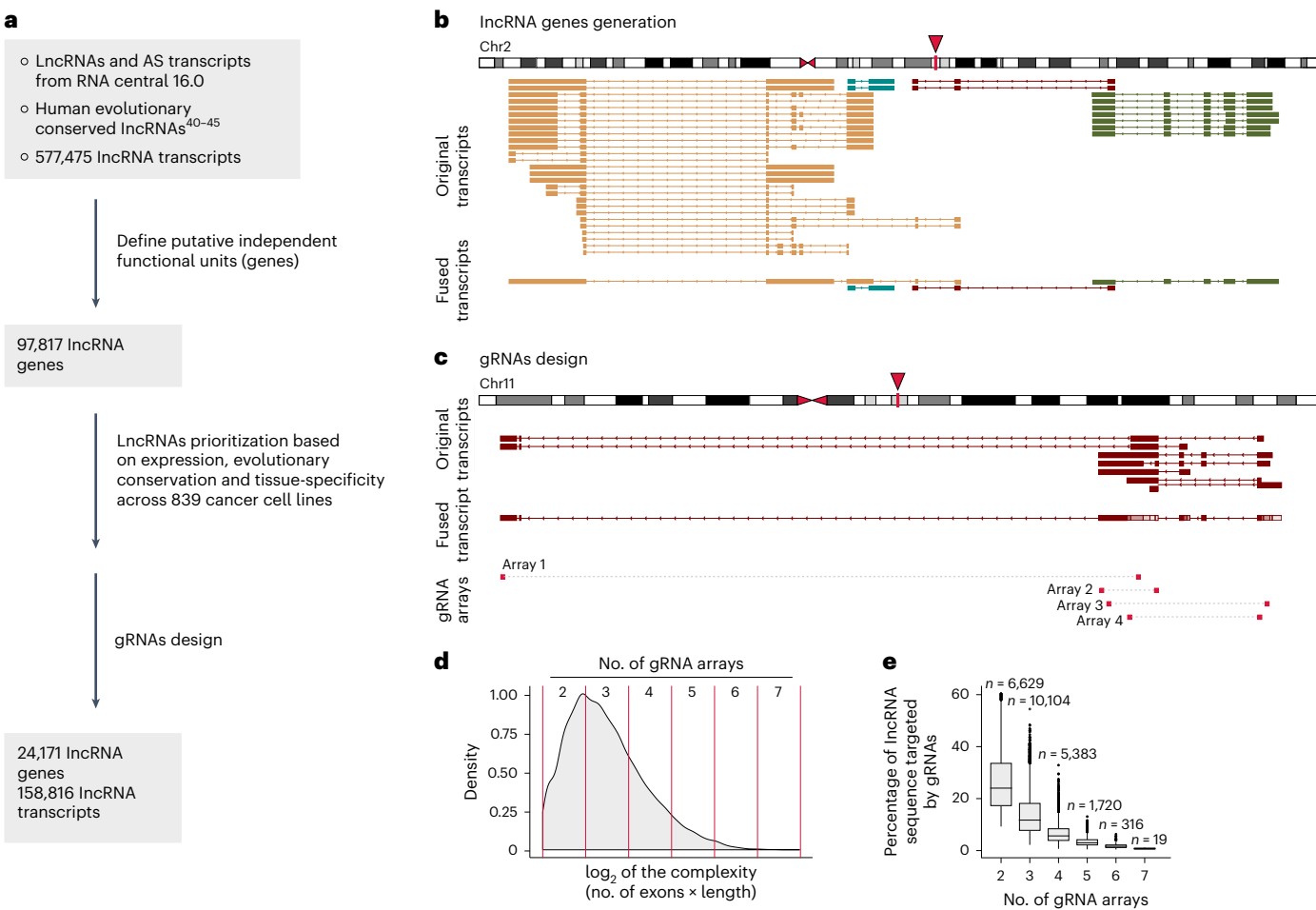

**Fig. 2 | CasRx pan-cancer library design. a**, Schematic workflow for selection of lncRNA genes to be targeted by our CasRx library. RNAcentral 16, https://ftp.ebi.ac.uk/pub/databases/RNAcentral/releases/16.0/. **b**, Example of similar lncRNA transcripts clustering into genes in a complex region of chromosome 2. Each putative independent functional unit is designated by a unique color, designated as an lncRNA isoform gene. **c**, Exemplary gRNA array design for a lncRNA gene composed of seven isoforms located on chromosome 11. **d**, Density plot displaying the complexity (number of exons multiplied by the gene length) of all 97,817 lncRNA genes. Values at the top indicate the number of gRNA arrays designed for lncRNA genes with different levels of complexity. **e**, Boxplot showing the gRNA sequence coverage (percentage of lncRNA sequence targeted by its specific gRNAs). LncRNA genes with extremely low complexity, characterized by an average of 1.4 exons and 573.3 bp in length, are targeted by only two arrays. For these genes, the gRNA sequence coverage is very high, making the design of additional arrays largely unfeasible. Boxplot data are presented as follows: center, median; box bounds, 25% and 75% percentile; whisker, 1.5× IQR. Outliers are marked as independent dots. *n*, number of lncRNAs for each category.

## CasRx library design

Most selected lncRNA genes have several isoforms and extensive structural heterogeneity, ranging from small 200 base pair (bp) single-exon lncRNAs to genes composed of 148 exons and 150 kb in length. We took several design decisions that account for these characteristics, thereby aiming to achieve high detection capacity while limiting library complexity/size (which compromises screening feasibility and throughput).

First, we exploited the capacity of CasRx to mature its own gRNA array[34] and designed each vector to encode a pair of nonredundant gRNAs targeting the same lncRNA gene. This 'dual targeting' facilitates effective transcript inactivation. Moreover, resulting fragments lack either the 5′-RNA cap or the poly-A tail, which are essential for RNA stability[53]. Second, we scaled the number of gRNAs according to lncRNA complexity (as defined by lncRNA lengths multiplied by the number of exons) by selecting between two and seven gRNA arrays (4–14 gRNAs) for each gene (Fig. 2d,e). Third, we developed an iterative gRNA design process, with each cycle having less strict requirements for gRNA quality[43] and gRNA distribution along each lncRNA gene (as a proxy to capture isoform complexity). Iterations were repeated until design criteria were fulfilled (Fig. 2a,c). To avoid possible off-targets, we discarded gRNAs with at least 23 nucleotide homology

to any coding or nonprotein-coding exonic sequence (23 nucleotides are necessary for optimum CasRx activity[43]). Based on these design considerations, we designed gRNAs predicted to target 24,171 lncRNA genes (Fig. 2a).

## Quality control of selected lncRNAs

We first analyzed the expression of our 24,171 lncRNA targets across cancers (Supplementary Fig. 3a). When using an expression threshold of >0.01 transcripts per million (TPM), which discriminates expressed from nonexpressed lncRNAs (Supplementary Fig. 3b), we found an average of 12,510 Albarossa-targeted lncRNA genes to be expressed in each of the 839 cancer cell lines (minimum, 7,934; maximum, 15,824) (Fig. 3a). This supports the broad applicability of the library. Since robustly expressed lncRNAs are potentially more relevant, we also explored data using expression thresholds of >0.1 TPM or >1 TPM. The average number of expressed lncRNA genes targeted by Albarossa at these thresholds was 8,192 and 2,357, respectively (Supplementary Fig. 3c).

Of note, despite this high 'cross-cancer coverage', the library captures a large part of the interentity lncRNA transcriptional heterogeneity, as shown by the enrichment of lncRNAs that are differentially

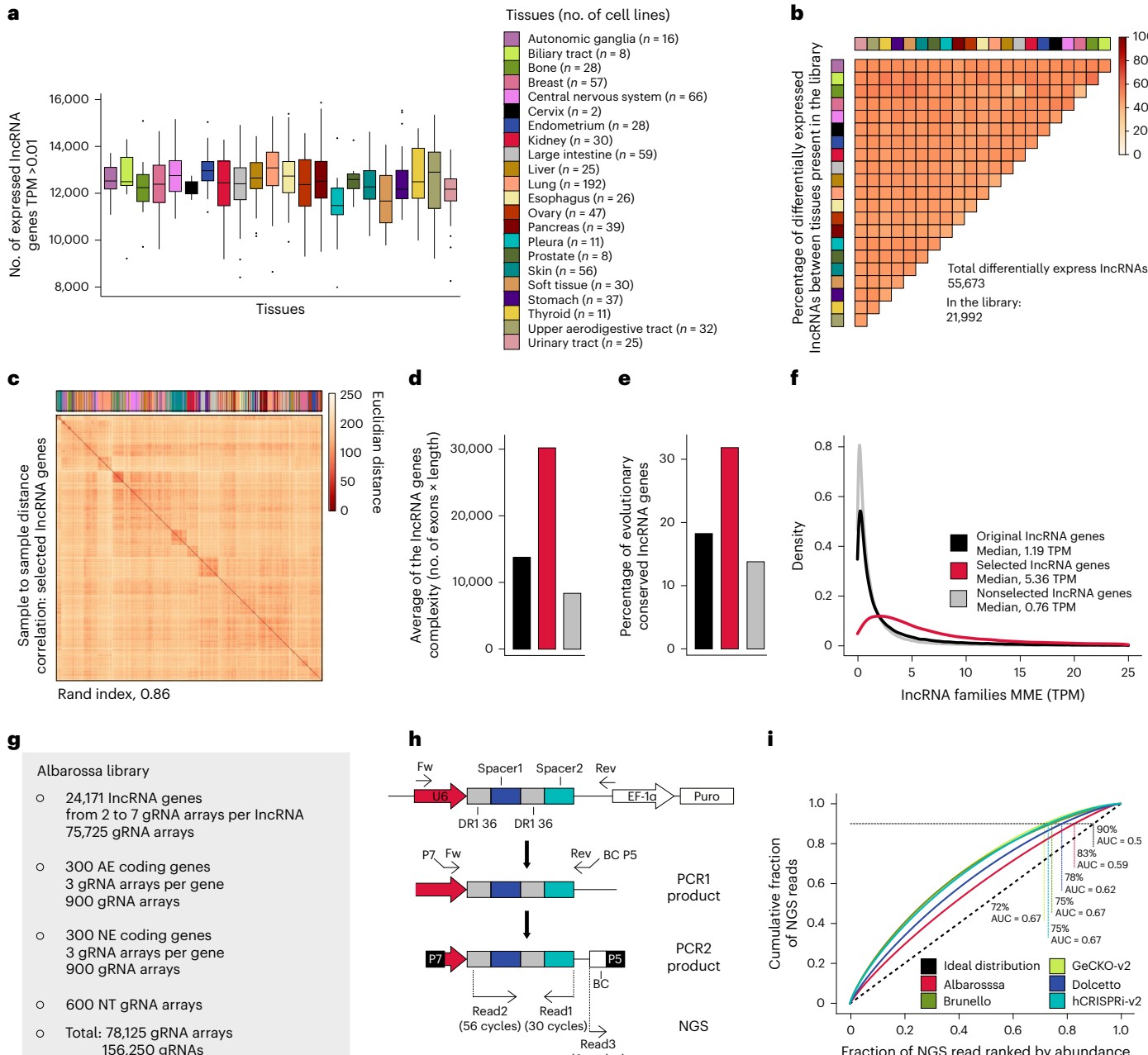

**Fig. 3 | Albarossa library displays high cross-cancer coverage and captures interentity lncRNA transcriptional heterogeneity. a**, Boxplot showing the number of expressed lncRNA genes represented in the library across cancer cell lines. LncRNAs with TPM >0.01 were considered and cell lines were grouped according to the tumor tissue of origin. Color legend in the right panel. Boxplot data are presented as follows: center, median; box bounds, 25% and 75% percentile; whisker, 1.5× IQR. Outliers are marked as independent dots; *n*, number of independent cell lines per tissue type. **b**, Heatmap representing the percentage of differentially expressed lncRNA genes between cancer cell lines from different tissues represented in the Albarossa library. Color code as in **a**. **c**, Sample-to-sample hierarchical clustering of cancer cell lines based on RNA-seq expression values from lncRNA genes represented in the Albarossa library. Rand index was calculated by comparing the clustering against the ideal tissue clustering. Color code as in **a**. **d**,**e**, Barplots displaying the average lncRNA gene complexity (**d**) or the percentage of evolutionary conserved lncRNA genes (**e**) in the original (black, all 97,817 lncRNA genes), selected (red, 24,171 lncRNA genes represented in the Albarossa library) or nonselected (gray) sets of lncRNA genes. **f**, Density plot showing the maximum median expression (MME) in the three indicated lncRNA groups. MME was calculated using the maximum expression value for each lncRNA gene across CCLE lines. **g**, Characteristics of the Albarossa library. **h**, Schematic of the NGS protocol developed for sequencing of the Albarossa library and the screened samples. **i**, Lorenz curves of the cumulative gRNA distribution in indicated CRISPR libraries. The dotted black line indicates the ideal distribution. Percentages indicate library representation at 90% of cumulative reads. BC, barcode; Fw, forward primer; Rev, reverse primer.

expressed between cancer types (Fig. 3b), and the clustering of cell lines by tissue type in sample-to-sample distance analyses (Fig. 3c and Supplementary Fig. 3d).

In a second step, we examined the representation of lncRNA characteristics indicating functionality and biological importance.

We found that lncRNA complexity is increased in our library (Fig. 3d). Indeed longer lncRNAs have a higher propensity form structured/functional regions[54]. In addition, our library displays a 2.3-fold enrichment with evolutionarily conserved lncRNAs, indicating selection across species (Fig. 3e). Finally, the median expression of lncRNAs

selected for our library is 7.1 times higher as compared with nonselected lncRNAs (Fig. 3f).

Thus, our lncRNA selection is strongly enriched with functional traits and displays high cross-cancer coverage while still capturing tissue heterogeneity.

## Library generation and sequencing

For library generation, we complemented the lncRNA targets (24,171 lncRNA genes) with positive and negative controls. We designed gRNAs against 300 AE and 300 NE protein-coding genes and, in addition, included 600 NT gRNA arrays (Fig. 3g). The library, which we refer to as Albarossa, comprises 78,125 gRNA arrays (156,266 gRNAs).

For cost-effective detection of both gRNA sequences present in individual arrays, we developed a paired-end next-generation sequencing (NGS) protocol based on a two-step nested PCR reaction. We used custom NGS primers to initiate sequencing close to the gRNA spacers. Read 1 starts directly at spacer 2, which is highly diverse DNA, supporting optimal cluster identification without PhiX spike-in. The use of a 75-cycle Illumina cartridge allows recovery of both spacers and the barcode (Fig. 3h) at low cost. We retrieved gRNA counts using a custom-made analysis script.

An even gRNA distribution in the library allows coverage reduction (cells per sgRNA)[55], thereby diminishing costs and experimental efforts. Figure 3i shows that gRNA representation is more even for Albarossa than for widely used genome-wide CRISPR libraries.

## Genome-scale CasRx screening

To map lncRNA dependencies, we used Albarossa for pooled fitness screens, which rely on CRISPR perturbation followed by capture of gRNA enrichment or depletion trends over time. We performed screens in glioblastoma (cell lines LN-18, LN-119), lung nonsmall cell cancer (A549, NCI-H460) and pancreatic ductal adenocarcinoma (KP-4, MIA PaCa-2). Control experiments without CasRx expression were conducted in one cell line per entity. For each screen, we transduced 94 million cells with the lentiviral library at a multiplicity of infection (MOI) of 0.25 (supporting transduction of one gRNA array per cell) and a coverage of 300. After puromycin selection, two replicates (20 million cells each) of individual lines were cultured for 21 days (Fig. 4a).

We determined gRNA frequencies for each screen by sequencing (Fig. 3h), yielding 15–20 million mapped counts per sample (Supplementary Fig. 4a,b). LFC values were calculated by comparing gRNA frequencies in screened samples and the original library. Technical replicate correlations (Supplementary Fig. 4c) were comparable with those of other loss-of-function lncRNA screens[15,16,27] (Supplementary Fig. 4d,e). As expected, positive control gRNAs targeting common essential protein-coding genes were depleted, whereas LFC values of negative control gRNAs (NT gRNAs or NE-gRNAs) remained around zero (Fig. 4b,c and Supplementary Fig. 5a,b). In experiments without CasRx activity, LFCs for positive and negative controls were centered around zero (Supplementary Fig. 5c).

To examine the sensitivity and specificity of our CasRx-based perturbation approach, we generated receiver operating characteristic curves using controls. Area under the curve (AUC) values for our screens ranged between 0.84 and 0.97, which are higher than reported for siRNA screens and slightly smaller than for Cas9 screens[56] (Supplementary Fig. 5d), confirming robust detection capability of our screens across lines. For some AE genes, fitness defects rely on full inactivation, which explains some differences between CasRx and Cas9-based perturbation.

Importantly, the overlap of gRNA LFC distribution curves targeting negative controls, nonexpressed or expressed lncRNAs supports the lack of indiscriminate off-target cleavage (Fig. 4d and Supplementary Fig. 5e).

## Systematic mapping of lncRNA dependencies in cancer

To identify lncRNA hits with fitness effects, we used the robust rank aggregation (RRA) method (from MaGeCK v.0.5.9 (ref. 57); Fig. 4e and Supplementary Fig. 6a). A fundamental difference between loss-of-function screens targeting lncRNAs and such targeting protein-coding genes, is the considerably higher number of true positive hits in the latter. Consequently, potential off-targets are more problematic for lncRNA screens than for protein-coding screens. To address this issue, we added another layer of filtering that 'blacklists' gRNAs, which map to common essential protein-coding genes when sequence homology criteria are being relaxed (23 nucleotide match, two mismatches allowed, exonic and intronic sequence considered). After removal of these 'blacklisted' gRNAs, we identified an average of 46 (26–77) lncRNA hits per cell line, amounting to a total of 206 hits (Fig. 4f). By comparing these screening hits with hits in control conditions without CasRx (false positives, n = 3) (Supplementary Fig. 6b) we calculated an empirical false discovery rate (FDR) of 0.022, thus supporting the specificity of the screen.

As another quality control, we analyzed the behavior of different gRNA arrays targeting the same lncRNA gene. We found that, for lncRNA hits, most arrays contribute to the fitness defects (Supplementary Fig. 6c), supporting the functionality of the array design.

To confirm enrichment of Albarossa with functional lncRNAs, we designed, for LN-18 cells, a 'Δ-library' targeting expressed lncRNAs that were not included in Albarossa. A screen using this library identified 26 lncRNA hits (Fig. 4g and Supplementary Fig. 6d,e). Subsequently, we calculated the discovery rate for both libraries (percentage of hits among targeted and expressed lncRNA genes). The discovery rate of Albarossa was two or three times higher for all hits or high-confidence hits, respectively (Fig. 4h). Thus, by targeting only one-quarter of all lncRNAs, Albarossa detected two-thirds of all and three-quarters of high-confidence lncRNA vulnerabilities. We therefore conclude that our approach reconciles high discovery power and broad entity representation with scalable experimental throughput for pan-cancer applications.

## CasRx-based screens overcome previous approaches limitations

In contrast to RNAi, CasRx is expected to be functional in the nucleus. To examine this, we performed cytoplasmic/nuclear fractionation followed by deep full-length RNA sequencing (RNA-seq) for all CasRx clones. We found an enrichment of screening hits in both the nuclear

**Fig. 4 | Genome-scale CasRx mapping of lncRNA dependencies across different tumor types. a**, Experimental outline of CasRx-based genome-scale lncRNA fitness screening developed in this study. **b**, Scatterplots of gRNAs LFC between two technical screening replicates. LFC values were calculated using CPM-normalized counts, with the plasmid library serving as a baseline. Three representative screens using the cell lines indicated are shown. gRNAs targeting lncRNAs are marked gray, negative control gRNAs (gRNAs targeting NE coding genes and NT gRNAs) are blue, and positive controls gRNAs (gRNAs targeting AE coding genes) are green. **c**, LFC distribution of negative and positive control gRNAs in indicated screens. **d**, LFC distribution of negative control gRNAs and gRNAs targeting expressed or nonexpressed lncRNA genes. **e**, Scatterplots displaying RRA scores of screened genes, as calculated by MAGeCK. NE coding genes are marked blue, and significant hits are marked red (lncRNA hits) or green (AE coding gene hits). Remaining genes are marked gray. Selected lncRNA hits are annotated. **f**, Summary table of significant hits in all CasRx screens. **g**, Scatter plot displaying RRA, as calculated by MAGeCK, for genes screened using a LN-18 Δ-library. The Δ-library targets all lncRNA genes expressed in the LN-18 cell line (TPM > 0.01) that are not targeted by the Albarossa library. Color scheme as in **e**. **h**, Barplots displaying the percentage of hits (left) or high-confidence hits (FDR < 0.05, right) among expressed lncRNA genes in LN-18 screens deploying the Albarossa library (red) or Δ-library (gray).

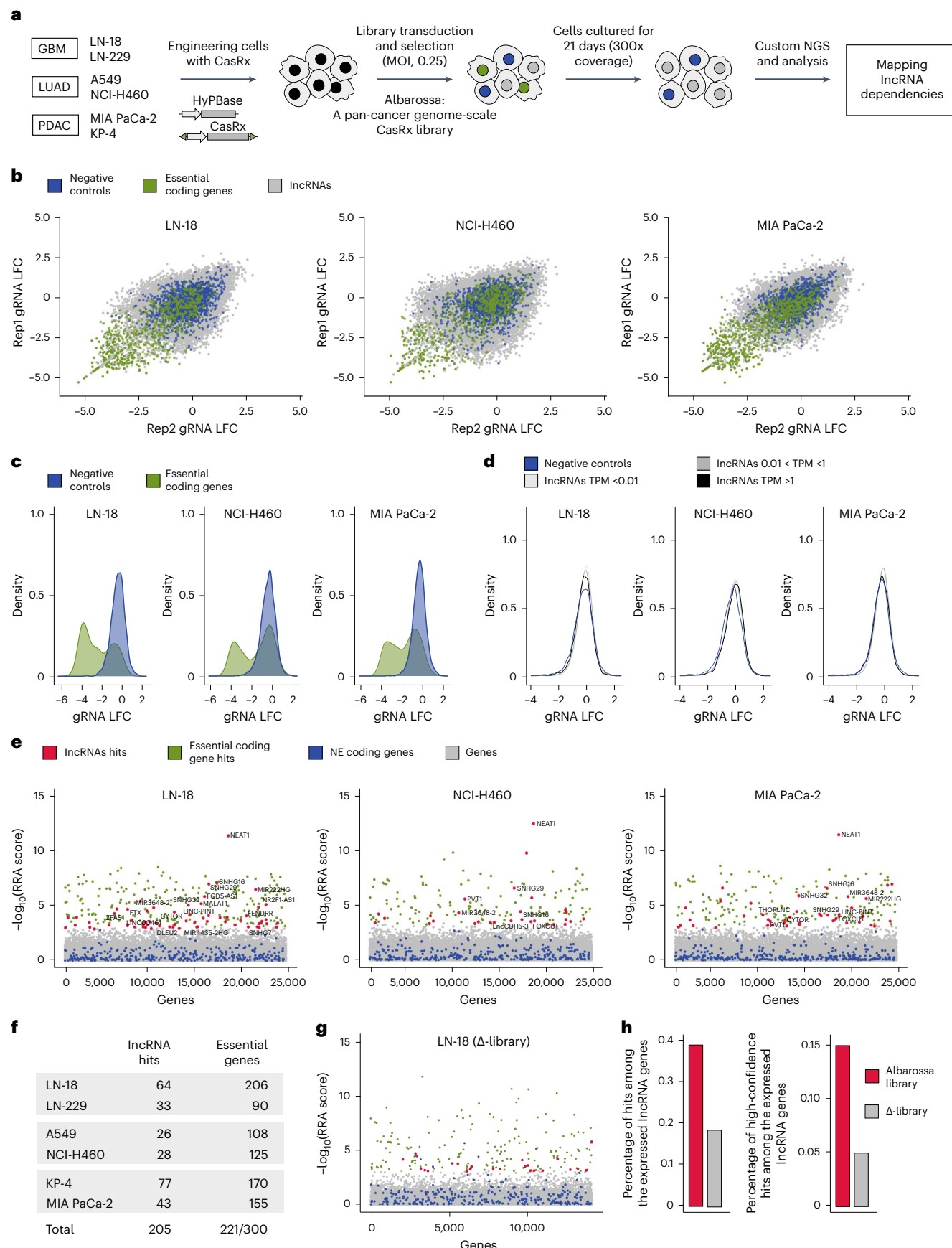

and the cytoplasmic fractions (Fig. 5a), showing detection of both types of lncRNA essentialities.

The quality of our screens is supported by comparisons to (d) Cas9-based lncRNA perturbation methods[3–5]. By reanalyzing the most comprehensive lncRNA screen so far, which is based on CRISPRi[27], we found several differences. First, to compensate for functional heterogeneity, the CRISPRi library has ten gRNAs per lncRNA. As a consequence, it is 2.1 times larger than Albarossa, but targets 1.5 times fewer lncRNAs (Supplementary Fig. 7a,b). Second, in contrast to CasRx, CRISPRi hits were strongly enriched for lncRNAs close to essential coding genes (Fig. 5b and Supplementary Fig. 7c), leading to large numbers of off-target hits that had to be removed[27] (Fig. 5c and Supplementary Fig. 7d–f). Third, although CRISPRi libraries were designed individually for each cell line, the screens identified fewer lncRNA dependencies than CasRx screens, which build on the 'pan-cancer' Albarossa library (Fig. 5d and Supplementary Fig. 7g,h).

Another study used Cas9-based deletion to screen for lncRNA dependencies[15]. Application of the same statistical approach for data analyses as for our CasRx screen revealed between zero and four hits per cell line (Fig. 5c). The low number of hits in that study might be related to the smaller library used and the low efficiency of Cas9-based generation of chromosomal deletions. Finally, when reanalyzing data derived from a screen targeting Cas9 splicing sites[16], we observed—in addition to the already reported issues related to false positives[26]—that the distribution of LFCs is shifted to negative values for all gRNAs (including such targeting nonexpressed lncRNAs) (Supplementary Fig. 7i). This known Cas9-related 'toxicity,' which is caused by double-strand cleavage[17–25], was not controlled for. Hence, the data structure does not support statistical analyses for comparisons to our screens.

We conclude that CasRx offers a powerful alternative to other lncRNA screening approaches. The strengths and advantages of our system relate to various screening aspects, including hit discovery efficiency, specificity, scalability and broad library applicability.

## Identification of common and context-specific vulnerabilities

To compare vulnerability profiles systematically across cell lines, for each significant hit we calculated a lncRNA-specific LFC value. We used these values for clustering based on occurrence across cell lines and found that 53.9%, 38.8% and 7.3% of hits were dropouts in one (cell-line-specific dependency), two to five (common-essentiality) or all cell lines (core-essentiality), respectively (Fig. 5e). We found that expression of hits is, on average, 4.5–8.7 times higher than of 'non-hit' lncRNAs, supporting functionality (Fig. 5f). A systematic literature search (Methods) revealed that 29.8% of our lncRNA hits had already been associated with cancer. Moreover, for 24.4% of hits, orthogonal lncRNA perturbation approaches identified a link to cellular proliferation. Of note, however, 70.2% of lncRNAs have not yet been associated with cancer or proliferation (Fig. 5e).

Among the 'core lncRNA vulnerabilities' are well-known structural lncRNAs[58–61], such as NEAT1 and MALAT1, as well as two lncRNA groups that are processed to small RNAs. This includes the small nucleolar RNA host genes SNHG16, SNHG29, SNHG32, SNHG6 and ZFAS1 and the miRNA host genes MIR222H and MIR3648-2. Although some members of these genes have been associated with cancer before, others are new, including SNGH32 and MIR348-2.

The 'common essential' group also encloses miRNAs host genes, including MIR22HG, MIR4435-2HG and MIR30DHG. Moreover, LINC-PINT—a well-studied tumor suppressive lncRNA gene[62,63] that can exert tumorigenic effects[64] in some cases through isoforms hosting pro-oncogenic miRNAs[65]—lies in this group. Other examples of lncRNA dependencies observed in several cell lines are SNHG7 and SNHG5 (SNHGs with oncogenic function), CYTOR[66,67] (involved in cell migration and cytoskeleton organization), PVT1 (ref. 68) (a MYC regulator, commonly overexpressed in cancer) or FTX[69,70] (a XIST regulator involved in DNA methylation and X chromosome inactivation).

As expected, most of our lncRNA hits are cell-line specific, confirming extensive context-dependency of lncRNA functionality. Several lncRNAs in this group, such as LINC00461, DLEU2, LINC00511, DLGAP1-AS1 and THORLNC (Fig. 5e), have been described previously to be oncogenic. Importantly, these lncRNAs display higher expression in cell lines in which they are a hit as compared with lines in which they are not causing fitness defects, supporting the notion that the screens recover biologically relevant targets (Fig. 5g).

## Validation of lncRNA dependencies

To further validate our screening results, we first designed a library comprising new arrays targeting 93 of our previously identified hits. We performed screens for each entity and found highly significant correlations to the previous screens (Fig. 6a).

Second, we performed fluorescence-activated cell sorting (FACS)-based competition assays for 16 lncRNA hits from our initial screens, including lncRNAs enriched in the nuclear (MALAT1, LINC-PINT, LINC00824, lnc-UTRN-3) and cytoplasmic (CYTOR, SNHG16) fractions. These experiments validated both common and specific lncRNA vulnerabilities (Fig. 6b and Supplementary Fig. 6e).

Third, we targeted six lncRNAs with alternative sgRNAs and performed clonogenic assays. As expected, targeting core-essential lncRNAs (NEAT1, SNHG32) reduced the fitness across cell lines, whereas common essential (MIR222HG and lnc-AKAP12-1) and cell-line-specific lncRNAs (FENDRR and HSALNG0116362) had effects only in specific cell lines, as predicted by the screen (Fig. 6c–e). We chose FENDRR for these analyses because it had been described previously as a tumor suppressor[71,72], but was predicted by our screen and confirmed to exert opposite effects.

Finally, we complemented these 'end-point' assays with serial quantification of proliferation over time (Fig. 6f). Overall, the results were in line with our observations made in the colony formation assays.

---

**Fig. 5 | CasRx-based screens solve bottlenecks linked to previous perturbation approaches and enable the identification of common and context-specific lncRNA vulnerabilities. a**, Cytoplasmic/nuclear expression LFC distributions of coding genes (gray) or for the indicated sets of lncRNAs (red/pink). The distributions represent average values for all cell lines used in this study. Genes with an LFC ≤ 0.75 or >0.75 were categorized as either nuclear-enriched or cytoplasm-enriched, respectively. **b**, Distance distributions of indicated sets of lncRNAs to the closest essential coding gene. Data represent screens based on CasRx (this study) and CRISPRi[27]. Two-sided Wilcoxon–Mann–Whitney test was used for statistical analyses. **c**, Boxplots showing the percentage of off-targets (left) or the percentage of off-target among the top hits (first quartile of the significant hits, based on FDR, right) per cell line. Data for our CasRx screens (red) and CRISPRi[27] screens (black) are shown; *n*, number of independent cell lines. **d**, Boxplot showing the number of lncRNA hits per cell line in the CasRx (red), CRISPRi[27] (black) or Cas9-based deletion[15] (gray) screens;

*n*, number of independent cell lines. **e**, Percentage of CasRx screening hits that have never been associated with cancer or proliferation before (black), have been associated with cancer (light gray) or whose fitness effects have been validated (light gray). The heatmap indicates unified lncRNA-specific LFC values for all lncRNA hits in the CasRx screens. Hits are clustered based on occurrence across cell lines (right panel). **f**, Boxplots showing the average expression of hit (red) or non-hit (gray) lncRNA genes per cell line. Lines connect the values for the same cell line; *n* = 6 independent cell lines. **g**, Boxplots showing the average expression of cell-line-specific hits as displayed in **e**. Each dot in the red box represents the average expression of lncRNAs in the specific cell line in which they are a hit. The connected dot in the gray box represents the average expression of the equivalent set of lncRNAs across the other cell lines; *n* = 6 independent cell lines. For all boxplots, data are presented as follows: center, median; box bounds, 25% and 75% percentile; whisker, 1.5× of IQR. NS, not significant.

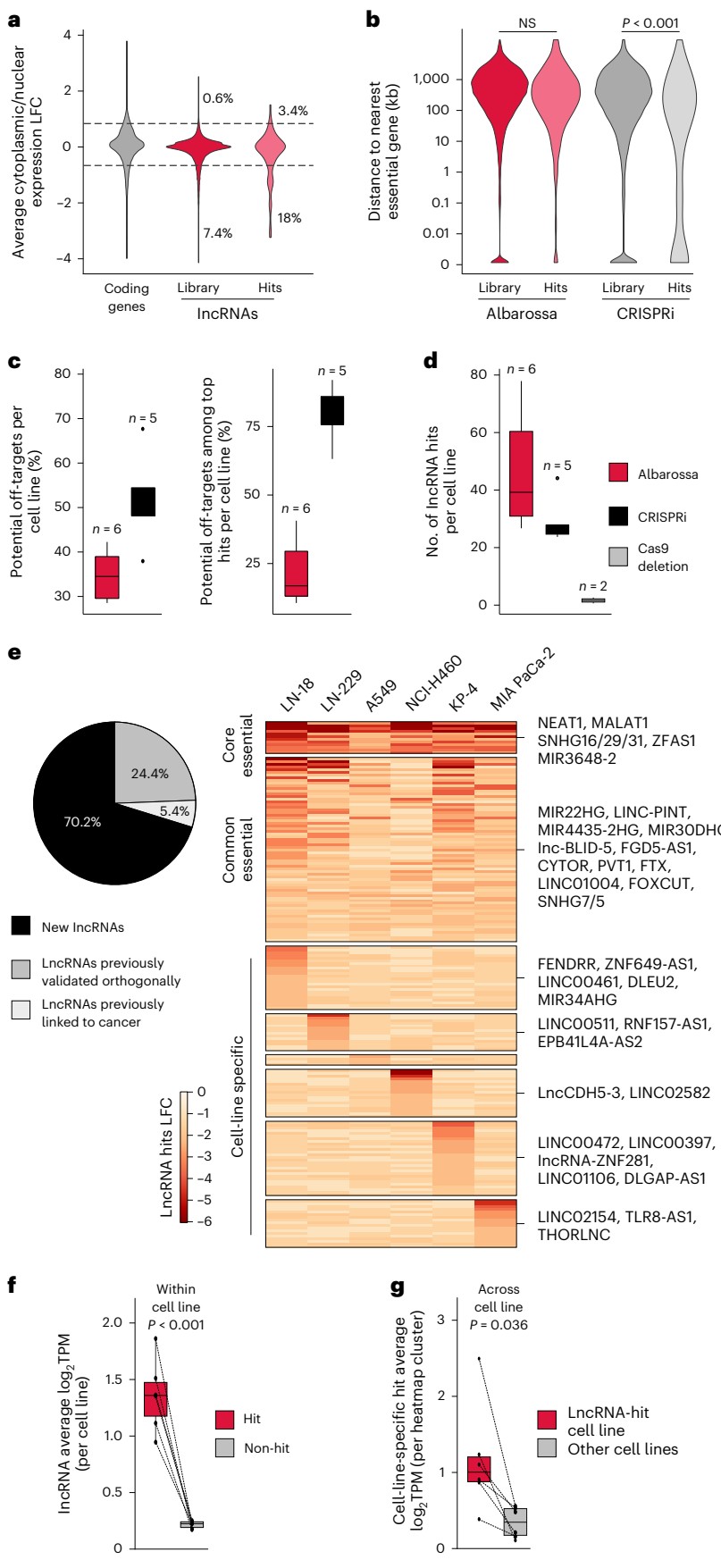

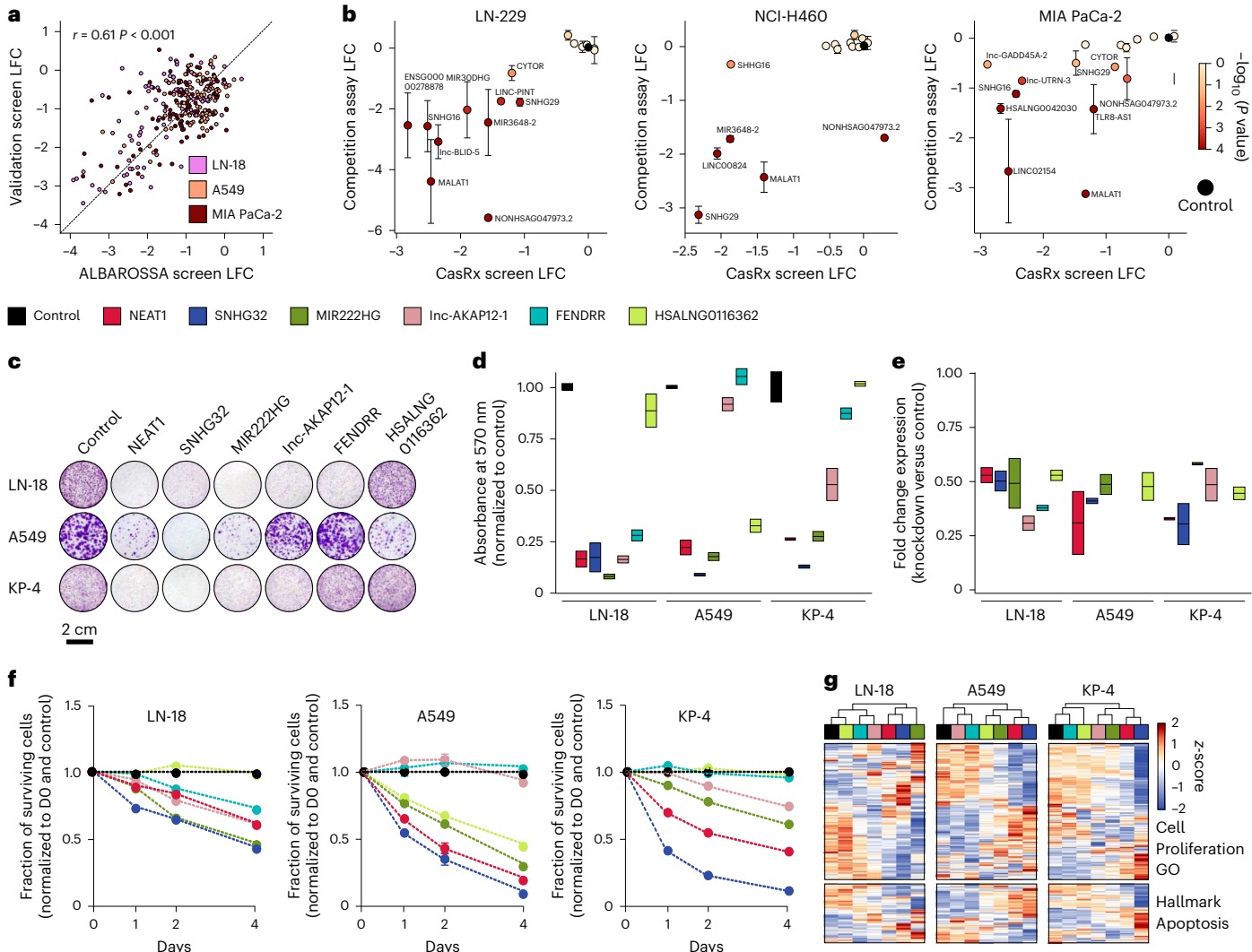

**Fig. 6 | Validation of lncRNA dependencies. a**, Screening-based high-throughput validation of lncRNA dependencies. Scatterplots display LFC values derived from screens performed using the Albarossa library and a 'validation library.' Correlation value and significance was calculated using the Pearson regression method. **b**, Individual hit validation using FACS-based competition assays in different cell lines. Scatterplots display LFC values for Albarossa screens and competition assays for indicated lncRNAs. LFC values represent the proportion of GFP-positive cells at day 14 versus day 0. GPF-positive cells transduced with NT gRNA served as controls. One-way analysis of variance with Dunnett's posttest was used to compare experimental and control conditions. Error bars, s.e.m.; $n = 4$ technical replicates from two independent experiments. **c**, Validation of fitness effects linked to perturbation of indicated lncRNAs using colony formation assays. Representative images are shown for different cell lines. Cells were transduced with a vector encoding a NT gRNA or gRNAs against different lncRNAs. After 2 days of puromycin selection, cells were seeded at low confluence and stained with crystal violet after 10 days of culture.

**d**, Quantification of colony formation assays. Colors indicate the different lncRNAs targeted; $n = 2$ independent experiments. **e**, qPCR-based quantification of knockdown levels for indicated lncRNAs. Data represent comparisons of knockdown and control conditions based on glyceraldehyde 3-phosphate dehydrogenase-normalized expression values; $n = 2$ independent experiments. For the boxplots, data are presented as follows: center, median; box bounds, minimum and maximum value. **f**, Proliferation curves indicating fitness effects linked to perturbation of indicated lncRNAs over time. LncRNAs were targeted as described in **c**. Cell numbers were normalized to day 0 (D0) and are indicated as fold change over controls (cells transduced with an NT gRNA); $n = 2$ independent experiments. **g**, A fraction of cells from **f** were collected at day 0 and processed for RNA-seq. The heatmaps show the expression pattern of genes (related to cell proliferation or apoptosis) that are significantly differentially expressed in at least one experimental condition against the corresponding control condition.

Likewise, RNA-seq early after sgRNA transduction/selection confirmed transcriptional changes for gene sets associated with proliferation and apoptosis in expected cell lines (Fig. 6g).

## Discussion

We developed a Cas13/CasRx-based screening platform for genome-scale interrogation of the lncRNAome. The methodology overcomes limitations of other lncRNA perturbation approaches related to targeting efficiency, off-targets and collateral perturbation of other DNA elements. We evolved the system toward pan-cancer applicability, which will facilitate systematic exploration of lncRNA biology, identification of related context-specificities and discovery of lncRNA vulnerabilities for therapeutic targeting.

A critical requirement was to ascertain high nuclease expression, which is required for continuous RNA degradation. We achieved this through transposon-based multicopy delivery of 'CAG-CasRx' expression cassettes to the genome. Discovery of coding and noncoding dependencies reinforces the functionality of the system. The screening

setting thus achieves an optimal window of functionality—characterized by sufficiently high CasRx on-target activity but no indiscriminate off-target RNA degradation.

We show that CasRx-based lncRNA screening solves principal bottlenecks inherent to other perturbation approaches. The use of RNAi is limited not only by its low nuclear activity[33] but also by high off-target rates[32]. CasRx, which is fused to a nuclear localization signal (NLS), not only displays nuclear activity but also targets lncRNAs with high specificity (the optimum gRNA length is 20–23 nucleotides[34,35], whereas siRNA seeds are 2–7 nucleotides[73–76]).

A main confounding effect in Cas9–CRISPRi lncRNA screens is collateral perturbation of neighboring coding/noncoding DNA elements, which drastically limits the number of lncRNAs that can be interrogated. We show that CasRx does not suffer this limitation. As an illustration, 37% of lncRNAs targeted by Albarossa overlap with coding genes, and 16% have their TSS closer than 1 kb to the TSS of a coding gene—meaning that 43% are difficult to target specifically with Cas9–CRISPRi. Thus, CasRx substantially expands the space of the human lncRNAome that can be interrogated. Another source of false positive hits in Cas9-based screens is DNA repair-associated 'toxicity'[17–19,21,22]. This effect is not distributed equally across the genome but can be affected by chromatin structure[24,25] or DNA copy number status[17,18], and hence differs between cell lines. For screens interrogating the protein-coding genome such effects can be corrected statistically[20]. In Cas9-based lncRNAs screens, the magnitude of the 'toxicity-effect' is similar, but because gene essentialities are far less frequent for lncRNAs than for protein-coding genes (the ratio of true/false positive lncRNA hits is low), its statistical correction is a main hurdle. Direct RNA targeting solves this problem.

In silico studies addressing the incomplete survey and annotation of the human lncRNAome were critical prerequisites for a rational design of our screening strategy. By accessing different data sources, we assembled a comprehensive collection of human lncRNAs, mapped isoform complexity and tissue distribution of the lncRNA transcriptome, and defined putative functional units that require independent interrogation.

Large-scale cross-entity mapping of lncRNA vulnerabilities is relevant. For example, lncRNAs often display context-dependent effects. Even for the best studied lncRNAs, contradictory functions have been reported, depending on the cellular context[77–80]. This is particularly true in cancer, where genetic alterations rewire pathways and vulnerabilities landscapes. Moreover, predicting cellular function is much more difficult for lncRNAs than for protein-coding genes, for which information such as domain structure and gene homology are available. Pan-cancer mapping of lncRNA dependencies combined with multiomic and phenotypic data integration will therefore be a decisive step toward systematic context-based inference of lncRNA function.

Screening the entire lncRNA transcriptome across many cell lines is limited by the enormous size of required libraries/experiments. Creating cell-line-specific smaller libraries targeting only expressed lncRNAs can address this problem, but limits both experimental throughput (as libraries would need to be designed, purchased, cloned and tested for each line) and comparative analyses. Indeed, use of different libraries is the main source of batch effects in comparative screens[81]. Our size-reduced library addresses these problems. It was achieved through a target prioritization procedure that incorporates considerations on lncRNA functionality (expression level, evolutionary conservation) as well as context-specificity (tissue/cell-type). This allowed us to focus the screen on the biologically more relevant lncRNAs while maintaining broad representation. Further technical design decisions (gRNA-multiplexing, lncRNA complexity-adapted gRNA-coverage) helped to increase detection capacity whilst limiting library size.

By interrogating different cancer types, our screens identified larger numbers of lncRNA dependencies per cancer cell line than previous approaches, which used cell-line-specific libraries. The quality

of the method is further supported by the discovery of an unprecedented number of core and common lncRNA essentialities. This not only includes known structural lncRNAs, small nucleolar RNA host genes or onco-lncRNAs, but also numerous unknown lncRNAs, of which some have been annotated only recently.

Our work conceptualizes an experimental approach for genome-scale pan-cancer interrogation of lncRNA essentialities. The genetic tools, experimental protocols and bioinformatics pipelines described here will fuel screening efforts toward a pan-cancer lncRNA dependency atlas.

## Material availability
The main plasmids generated in this manuscript have been deposited in Addgene (Addgene cat. nos. 212961–212966 and 212972), https://www.addgene.org/browse/article/28243713.

### Reporting summary
Further information on research design is available in the Nature Portfolio Reporting Summary linked to this article.

## Online content

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

## Methods

### Cell culture, virus production and transfection

Cell lines (LN-18, LN-29, KP-4 and MIA PaCa-2) were cultured in DMEM-high glucose (Sigma) or (cat. nos. A549 and NCI-H460) with RPMI Medium 1640 plus L-glutamine (Gibco) in both cases supplemented with 10% FBS Superior (Sigma). Cell lines were tested regularly for mycoplasma and tested immediately before the beginning of the screens.

Lentiviral supernatants were produced in HEK-293T seed in 100 mm culture plates (Falcon) at 80% confluency. HEK cells were transfected with the packaging plasmids psPAX2 (1.25 μg), pMD2.G (0.75 μg) and with the appropriate lentiviral vector (2 μg) using TransIT-LT1 (Mirus) according to the manufacturer's protocol. The day after transfection, we added fresh medium to the cells and collected it the next day; 20% of total medium supplemented with 10 μg ml⁻¹ of polybrene (Merck) was used to transduce cell lines, previously seeded at 40% of confluency. In the case of the Albarossa library virus, cells were transduced using a spin-infection protocol: six million cells were resuspended in 4 ml of medium with the appropriate amount of virus, 10% FBS and 10 μg ml⁻¹ polybrene. The mixture was seeded in a six-well plate (Falcon) and centrifuged at 1,000$g$ for 2 h at 33 °C.

For the generation of CasRx-engineered cell lines, we first seeded the cells in a well of a six-well plate (Falcon) at 60–80% confluence. Cells were transfected with a mix of the EF-1a-HyPBAse and the PB-CAG-CasRx-Blast vector at a 1:5 molar ratio (2.24 μg in total) using Lipofectamine 3000 (ThermoFisher). The day after transfection, we added fresh medium to the cells; 2 days after transfection, cells were seeded with blasticidin and maintained in selection for 2 weeks. After selection, single cells were seeded by limiting dilution in 96-well plates (Falcon). Single clones were expanded and tested for CasRx activity.

### Quantitative transposon insertion site sequencing

Quantitative transposon insertion site sequencing (QiSeq) was performed as described previously[39,40]. Briefly, genomic DNA was sheared using a Covaris M220 sonicator, yielding 250 bp fragments. These fragments were subjected to end repair and A-tailing. Then, a splinkerette adapter was ligated at both ends of the fragmented DNA. Subsequently, the 5' and 3' transposon ends were processed separately using different end-specific primers. An initial PCR amplification step selectively amplified transposon-containing regions using transposon-specific primers in conjunction with splinkerette binding primers. The splinkerette is a Y-shaped double-stranded oligonucleotide with a template strand and a hairpin strand: in the initial PCR cycle, the splinkerette lacks a primer-binding site because the template strand exclusively contains the primer-binding site template, whereas the hairpin cannot serve as a primer-binding site. Consequently, only the transposon-specific primer functions in the first cycle, ensuring selective amplification of transposon-containing fragments. Subsequently, a second PCR step was employed to introduce sample-specific barcodes and Illumina P5 and P7 adapter sequences to the amplified fragments. This resulted in the generation of DNA fragments comprising the P7 adapter, the splinkerette, a segment of genomic DNA encompassing the insertion site, the transposon sequence and the P5 adapter sequence. Each sample was then quantified with real-time quantitative PCR (qPCR) (using P5-specific and P7-specific primers). Subsequently, samples were mixed equimolarly and the library pool was again quantified. Libraries were sequenced using the Illumina MiSeq sequencer in a paired-end configuration. Reads were subsequently aligned to the GRCh38 reference genome using bwa-mem algorithm[82], and only sequences containing transposon-genome junctions were quantified. Insertions with more than 20 counts were curated manually and insertion sites without transposon insertion site (TTAA), or that were detected in several samples (common artefactual insertions), were

classified as artifacts. Then, we calculated the normalized coverage per each insertion by dividing the counts of each insertion by the total sample count and then multiplying the result by 100. Insertions that were not categorized as artifacts and with a normalized coverage higher than two were considered reliable (Supplementary Table 1).

### Flow cytometry and CasRx activity quantification

CasRx-engineered cells were transduced with the pLenti-EGFPd-establized-Hygro (Addgene, cat. no. 138152 (ref. 43)) lentivirus and selected with hygromycin for 1 week. After selection, the cell culture was split into two different dishes. One of the dishes was transduced with the pLKO5-CasRx(DR1 30)–EFS-tRFP657 containing a NT gRNA and the other with a GPF-targeting gRNA. Four days after transduction, the cells were resuspended in FACS buffer (PBS, 3% FBS, 5 mM EDTA) and recorded using a Beckman Coulter Cytoflex LX system. FlowJo v.10.7.2 was used to analyze populations. CasRx activity was calculated as one minus the ratio between the GFP intensity (mean B525-FITC-A) of cells transduced with the gGFP gRNA and cells transduced with the gNT gRNA. The pLKO5-CasRx(DR1 30)–EFS-tRFP657 vector was cloned using the pLKO5.sgRNA.EFS.tRFP (Addgene cat. no. 57823 (ref. 83)).

### Assembly of a collection of human lncRNA transcripts and generation of lncRNA genes

Genomic coordinates of transcripts classified as lncRNAs or antisense were retrieved from RNAcentral *Homo sapiens* release 17.0. This collection was complemented with human evolutionary conserved lncRNA transcripts from selected publications[47–52]. To integrate both collections, the coordinates from refs. 47,49–52 were converted from Genome build hg19 to hg38 using liftover from University of California Santa Cruz[84]. The transcript coordinates for the genes in ref. 48 were retrieved from ENSEMBL release v.103 (ref. 85). In total, we collected 577,475 individual lncRNA isoforms (Supplementary Table 2).

RNAcentral is a comprehensive database that includes several isoforms per lncRNA locus, some of which have low abundance and might originate from RNA-processing errors such as intron-retention. We intended to remove such 'artifacts.' Since introns are generally longer than exons and the calculation of TPM considers gene length, we excluded nonexpressed exons from our annotation and ensure a fair calculation of the expression of each lncRNA gene. We downloaded raw FASTQ files from RNA-seq data from the cells in the Cancer Cell Line Encyclopedia (CCLE)[86] (SRA database[13], accession number PRJNA523380). Raw reads were processed with Trimmomatic v.0.39 (ref. 87) to remove Illumina Adapter sequences and leading and trailing bases with a Phred score lower than 25, keeping only reads with a minimum length of 50 bp after filtering. Trimmed reads were aligned to the GRCh38 human genome build using STAR v.2.7.5b (ref. 88). The resulting BAM files were merged together and used to retrieve the base-pair-wise read coverage for each of the exons using pileup from samtools v.1.10 (ref. 89). We excluded every exon in which 90% of its length was not covered by at least one read.

Following an iterative process using custom scripts along with bedtools[90], we classified the lncRNA species into independent putative functional units (lncRNA genes). First, isoforms annotated in the same strand were processed together. Second, transcripts were grouped if they fulfilled two requirements: the overlap of the transcript body was ≥60% (referred to the shortest transcript) and at least one exon between two transcripts shared ≥60% of the genomic coordinates of the longest exon. Third, monoexonic transcripts that shared ≥30% of their genomic localization with one of the previous groups were incorporated into the group. Fourth, multiexonic transcripts that shared ≥90% of their total exon coordinates with one of the previous groups were incorporated into the group. Finally, we avoided bias in expression quantification due to overlap with protein-coding genes and the design of gRNAs targeting protein-coding genes. For this reason, we removed from the annotation any lncRNA regions that intersected with exons of any protein-coding

gene (Gencode v.38 (ref. [10])) in any strand or lncRNA regions that overlap with introns of any protein-coding gene on the same strand. Transcripts that lost >70% of their genomic coordinates in this process were removed. After all the steps, we obtained 97,817 lncRNA genes and their custom coordinates (Supplementary Tables 3 and 4).

## LncRNA genes expression quantification

We utilized two distinct methods to quantify our lncRNA genes expression across CCLE cell lines:

(1) A first 'sensitive/permissive' approach (STAR[88]) in combination with featureCounts[91]), which served as the basis for selection of lncRNA genes for CRISPR library design. The key goal at this stage was to not 'lose' overlapping lncRNAs, for which accurate quantification is very difficult (and which are—because of this—prone to be filtered out by more stringent approaches). In this approach, we first mapped reads to the genome using STAR v.2.7.5b (as above in 'Assembly of a collection of human lncRNA transcripts and generation of lncRNA genes'). Subsequently, the mapped reads were fed to featureCounts (subread package v.1.6.3)[91] to quantify lncRNA expression. We allowed for multioverlap of features, ensuring that if a read aligns to a location encoding several features (overlap of more than one lncRNA gene), it is counted once for each feature. In contrast to protein-coding genes, this scenario is very common for lncRNAs. In this step, we excluded multimapping reads, thereby avoiding inclusion of reads aligning to several genomic positions, such as transposable elements (Supplementary Table 5).

(2) A second method for more 'stringent' quantification of lncRNA expression, which served as a basis for a 'fair' calculation of the number of expressed lncRNA per cell line. While featureCounts proved very useful for lncRNA selection to be considered for library design, we were aware of its potential to overestimate the number of expressed lncRNAs per cell line. Therefore, to be conservative in determining the number of lncRNAs expressed in each cell line and included in our library, we reanalyzed RNA-seq data from CCLE with Salmon v.1.5.2 (ref. [92]), using as reference our custom coordinates (Supplementary Table 3) together with the protein-coding transcripts coordinates (Gencode v.38 (ref. [10])). Salmon employs a sophisticated statistical approach to quantify overlapping regions[92], ensuring that expression is not overestimated (individual reads are not being double-counted). To prevent spurious read mapping, particularly from transcribed repetitive regions, we incorporated the full GRCh38 build[93] as a decoy (Supplementary Table 6).

For differential expression analysis of lncRNAs between tissues, we employed limma v.3.42.2 (ref. [94]) using quantile as normalization method. LncRNA genes with a TPM value below 0.01 were excluded from the analysis.

## RNA-seq analysis

Total RNA was extracted and purified with an RNeasy Mini Kit (Qiagen) according to the manufacturer's instructions. RNA was quantified using the QuBit v.2.0 RNA High Sensitivity kit (ThermoFisher Scientific). Nuclear/cytoplasmic fractions were prepared using the nuclear/cytosol fractionation kit from Abcam (cat. no. ab289882) and RNA extracts were prepared as above.

The total, nuclear and cytoplasmic RNA from the CasRx clones or the total RNA from the parental cell lines (Fig. 5a) was sent for sequencing to Novogene. mRNA was purified using poly-T oligonucleotide-attached magnetic beads. Following fragmentation, first-strand cDNA was synthesized with random hexamer primers, followed by second-strand cDNA synthesis. The library was subjected to end repair, A-tailing, adapter ligation and size selection. After amplification and purification, the insert size of the library library was verified using an Agilent 2100 bioanalyzer and quantified via qPCR. Subsequently, the libraries were sequenced on an Illumina NovaSeq 6000 S4 flowcell with PE150 sequencing. Transcript quantification was performed using Salmon v.1.5.2 (ref. [92]), as detailed in the previous section (Supplementary Table 7). Subsequent analysis was conducted using R v.3.6.3. Specifically, principal component analysis (PCA) was conducted by selecting the top 10% of the most variable protein-coding genes based on their s.d. Differential expression analysis and Rlog count normalization were accomplished using DESeq2 v.1.26 (ref. [95]). The $\log_2$ fold change (LFC) of transcript abundance between the cytoplasmic and the nuclear fractions was calculated by dividing the cytoplasmic expression (counts per million (CPM) + 0.5) by the nuclear expression (CPM + 0.5) (Supplementary Table 7).

The RNA from the lncRNA dependencies validation (Fig. 6g), and from the indiscriminate off-target RNA cleavage experiments (Fig. 1h,i) was processed as follows: library preparation for bulk RNA-seq was done as described previously[96]. Briefly, barcoded cDNA of each sample was generated with a Maxima reverse transcriptase polymerase (ThermoFisher) using an oligonucleotide-dT primer containing barcodes, unique molecular identifiers (UMIs) and an adapter; 5′-ends of the cDNAs were extended by a template switch oligonucleotide and full-length cDNA was amplified with primers binding to the template switch oligonucleotide-site and the adapter. An NEB UltraII FS kit was used to fragment cDNA. After end repair and A-tailing, a TruSeq adapter was ligated and 3′-end fragments were finally amplified using primers with Illumina P5 and P7 overhangs. In comparison with ref. [96], the P5 and P7 sites were exchanged to allow sequencing of the cDNA in read1 and barcodes and UMIs in read2 to achieve better cluster recognition. The library was sequenced on a NextSeq 500 (Illumina) system with 65 cycles for the cDNA in read1 and 16 cycles for the barcodes and UMIs in read2. Data were processed using the published Drop-seq pipeline (v.1.0) to generate sample- and genewise UMI tables[97]. Reference genome GRCh38 (ref. [93]) was used for alignment. Transcript and gene definitions were used according to the Gencode v.38 (ref. [10]). Differential expression analysis was performed in R v.3.6.3 with DESeq2 v.1.26 (ref. [95]), testing each treatment against the respective control condition.

## Target selection and Albarossa library design

Using an iterative process, we selected 30,510 lncRNAs based on expression and evolutionary conservation from the original genes. The selection steps were performed on the calculated TPM values from the raw counts obtained with featureCounts subread package v.1.6.3[91] (Supplementary Table 5). We subsequently selected the 9,300 most expressed lncRNA genes on average in all the cell lines, the 9,306 most expressed genes on average in each tissue, the 8,892 most expressed genes by cell line and, finally, 3,012 evolutionary conserved lncRNA genes (2,000 with the maximum average expression across samples and 1,012 with the maximum tissue-specific expression).

To maximize the targeting of the most expressed isoforms we avoided designing gRNAs against low-expressed exons. We quantified exon expression levels (normalized by length) using featureCounts (subread package v.1.6.3)[91] with the -f option to output exon-level instead of transcript-level counts (Supplementary Table 8). We considered an exon as low-expressed if its normalized expression level was less than two s.d. from the mean normalized expression level of all the exons from the same lncRNA gene.

For each of the 30,510 selected lncRNA genes, we designed from two to seven crRNA arrays (each array containing two gRNAs). The number of arrays per gene was scaled according to the gene complexity (as defined by gene length multiplied by the number of exons). Using an in-house command line tool that allows parallel multiparametric gRNA design, we generated the arrays for each gene: (1) we retrieved

the FASTA sequence for the lncRNA genes using the getSeq function from the BSgenome package (*H. sapiens* GRCh38). (2) All possible 23 bp gRNA spacers and their predicted efficiency were generated using an adaptation of the pipeline used in ref. 43 (https://gitlab.com/sanjanalab/cas13). (3) To filter out possible gRNAs with potential off-targets, we mapped the gRNA sequences to all the genes from our lncRNA original genes and to all the protein-coding genes (Gencode v.38 (ref. 10)) using bowtie2 v.2.3.5.1 (ref. 98) end-to-end option without allowing mismatches. All the gRNAs with at least one off-target mapping were removed for the next steps. (4) We also removed gRNAs containing the restriction site sequence of the BsmBI enzyme (CGTCTC) to avoid interference during library cloning. (5) We extended the length of the gRNAs to 30 bp following the sequence of the targeted lncRNA. (6) We designed the arrays using four subsequent rounds of decreasing stringency (lncRNA genes whose design was not accomplished in the previous round were processed in the following round) according to three parameters: gRNA quality (Standardized Guide Score (SGS)[43]); a minimum distance between gRNAs in the same array (defined as minimum local distance (MLD)); a minimum distance between any gRNA targeting the same gene (defined as minimum global distance (MGD)). First, round, SGS ≥ 0.75; MLD ≥ 30% of the lncRNA gene length; MGD ≥ 10% of the lncRNA gene length. Second round, SGS in the best quartile; MLD ≥ 25% of the lncRNA gene length; MGD ≥ 5% of the lncRNA gene length. Third round, SGS in the best quartile; MLD ≥ 10% of the lncRNA gene length; MGD ≥ 1% of the lncRNA gene length. Fourth round, SGS any value; MLD≥ of 10% of the lncRNA gene length; MGD≥ no overlap. In each round, we selected the gRNA in position 1 for each array as follows: first, the gRNA with the best SGS was assigned to position 1 of the first array. For position 1 of the second array, we selected (of the remaining designed gRNAs) the one with the best SGS that fulfilled the MGD criteria. The same process was repeated for all the remaining arrays. We choose the gRNA in position 2 for each array as follows: for the first array, we picked (from the remaining gRNAs) the one with the best SGS that satisfied both the MGD and MLD criteria. This process was iterated for all arrays. If, at any point during this process, no gRNA meeting the SGS, MGD or MLD criteria was found, the lncRNA passed to the next less stringent design round. Based on the criteria described above, we designed gRNA arrays predicted to target 24,171 lncRNA genes.

We complemented the library with positive and negative controls. We designed gRNAs against 300 AE and 300 NE protein-coding genes. In addition, we included 600 NT gRNA arrays. AE and NE control genes were obtained from ref. 99. For these genes, we retrieved the ENSEMBL canonical transcript and we designed the gRNA arrays as described before for the lncRNAs. NT gRNA arrays were designed by first generating 10,000 random 23-mers. These sequences were mapped to the genome build GRCh38 (ref. 93) with bowtie v.1.2.3 (ref. 100) allowing up to three mismatches. Only sequences that did not map were retained. We selected 1,200 of these sequences and we extended their lengths to 30 bp using randomly generated seven-mers. Pairs of these gRNAs were used to design 600 NT arrays. The final library comprises 78,125 gRNA arrays (Supplementary Table 9).

## Design of other libraries

For the construction of the control library, NE coding genes were selected to do not show any fitness effect and to exhibit robust expression across the CCLE cell lines[45]. Specifically, we downloaded CRISPR–Cas9 and RNAi fitness effect data from the DepMap resource v.22Q2 (ref. 45). Genes with average and median fitness values between ±0.05 (Chronos fitness effect for CRISPR–Cas9 screens) and between ±0.15 (Demeter2 fitness effect for RNAi screens) were chosen. Next, we filtered out genes that were not expressed, excluding those with a median expression of <0.5 TPM across the CCLE cell lines. From the remaining genes, we manually selected 300 NE coding genes that displayed no observable fitness effect and exhibited expression across

CCLE cell lines. Additionally, we included 300 essential coding genes: we identified all genes showing a top 25% Chronos negative fitness effect in each specific cell line. Among these genes, we selected those appearing in this rank for all the cell lines and, subsequently, the genes with the highest average fitness effect were included in the library. For both the NE coding genes and the essential coding genes, four gRNA arrays were designed. We also included 1,634 NT gRNA arrays (Supplementary Table 9).

The LN-18 custom library targets all the lncRNA genes expressed in the LN-18 cell line (TPM > 0.01) and not targeted by the Albarossa library. The validation library targeted 93 lncRNA genes previously identified as hits in the Albarossa screens. Both libraries also included 1,200 gRNA arrays targeting nonessential coding genes, 1,200 gRNA arrays targeting essential coding genes and 1,000 NT gRNA arrays (Supplementary Table 9).

For the three libraries, gRNA arrays targeting genes and NT gRNA arrays were designed as described above in 'Target selection and Albarossa library design.'

## Cloning of CasRx libraries

An ssDNA oligonucleotide pool (TWIST Bioscience) containing all the sequences of the library arrays flanked by BsmbI (Supplementary Table 9) was amplified (three cycles) using the KAPA HiFi HotStart polymerase. The PCR reaction product was purified using the QIAquick PCR Purification kit (Qiagen) and the resulting product was cloned into the lentiGuide-CasRx(DR1 36)-stuffer-Puro (Supplementary Table 10) using several Golden Gate assembly reactions. The cloning reactions were pooled together and precipitated using isopropanol-sodium acetate and washed three times with 70% ethanol. The concentrated DNA was electroporated into competent *Escherichia coli* cells. The cells were cultured in Luria Bertani medium with (100 µg ml⁻¹) ampicillin selection overnight. The next day, plasmid DNA was purified using a Plasmid Mega kit (Qiagen). The quality of the library was checked by digestion, Sanger-sequencing and NGS (see below). To maintain library diversity, the coverage during cloning was around 6,000. The lentiGuide-CasRx(DR1 36)-stuffer-Puro vector were cloned using the lentiGuide-Puro (Addgene, cat. no. 52963 (ref. 101)).

## CasRx screens and NGS

For all the screens, 94 million cells were transduced (using the previously described spin-infection protocol) with the Albarossa lentiviral library at a MOI of 0.25 and a coverage of 300. On the following day, all the cells were seeded into 175 cm² flasks (Cellstar) with puromycin. After 4 days of selection, two replicates (20 million cells each, maintaining the 300 coverage) of individual lines were seeded into 150 mm culture plates (TPP). Each replicate was passaged every 3 days (maintaining the coverage) and cultured for 21 days. At the end of the experiment, 24 million cells were harvested and processed to isolate genomic DNA to be used for sequencing.

NGS libraries were produced following a two-step nested PCR protocol. In the first PCR (18 cycles), we enriched a 481 bp genomic region containing the integrated array using the KAPA HiFi HotStart polymerase (Roche) with 6 µg of DNA in each 50 µl reaction (25 reactions per sample). We pooled all reactions and purified the DNA from 100 µl using the QIAquick PCR Purification kit (Qiagen). We used the elute to amplify the final library in a second nested PCR (ten cycles) using the KAPA HiFi HotStart polymerase (Roche) and barcoded primers that contained the P5 and P7 Illumina Adapters. We quantified the library and sequenced it on an Illumina NextSeq 500/550 High Output Kit v.2.5 (75 cycles) with custom primers. We obtained three reads: in read1 with 30 cycles, the primer binds next to the second guide and gives the sequence in reverse complement. In read2, the primer binds to the passage of U6 to the direct repeat and reads the first guide in cycles 29–56; i7 barcodes (6 bp) are sequenced with a third primer.

## Screens analysis

Using a custom pipeline, we obtained the raw counts from all the screen samples and the original plasmid library (using the FASTQ files from the NGS protocol described above). We generated two different indexes for the sequence of the gRNAs in position 1 or position 2 of the arrays. For position 1 we indexed the first 27 bp (length of the gRNA covered by the sequencing procedure) and for position 2 the full 30 bp. The first 29 bp of Read2 were trimmed to leave only the 27 bp portion that covers the gRNA at position 1. We used bowtie v.1.2.2 (ref. 100) to map read1 to the position 2 index and read2 to the position 1 index allowing two mismatches. We assigned a count if the sequence from both read1 and read2 mapped to the same array.

For each gRNA array, we calculated an LFC on CPM-normalized counts using the plasmid library as the baseline. Arrays with an absolute LFC value in the percentile one in at least two control screens (without CasRx) and one sample screen were considered noisy and removed from the analysis. Arrays with <30 counts in the plasmid library were also removed. For the remaining arrays, we normalized the counts using the median of ratios method from DESeq2 (ref. 102). The P value of each array was calculated using the RRA algorithm from MAGeCK v.0.5.9 (ref. 57), treating the sequenced plasmid library as control. As a second criterion to prevent off-targets gRNAs, we 'blacklisted' any array with a significant P value (<0.05) in a minimum of two screens and in which at least one of its gRNAs mapped (using bowtie v.1.2 (ref. 100) allowing 21 nucleotide end-to-end) to the exon or intron of an essential coding gene[45,103,104] (Achilles common essentials and CRISPR common essentials lists from the DepMap repository version 22Q2). More precisely, we used a 23 nucleotide match allowing two mismatches at any position. This means by default that a 21 bp perfect match is included. By allowing two mismatches, we are more permissive in detecting off-targets as we allow for a larger potential 'sequence pool' to align and be flanked as potential off-target (Supplementary Table 11).

After using the RRA algorithm from MAGeCK v.0.5.9 (ref. 57) and the plasmid library as baseline, we retrieved an FDR for each gene in every screening. MAGeCK identifies a gene as a dependency if some gRNAs display a markedly strong depletion or if several gRNAs with moderate to strong effects consistently demonstrate the same behavior. It then evaluates the chance of a random set of gRNAs behaving similarly to those targeting a specific gene. The rarer the observed pattern is, the more significant the resulting P value becomes. A gene was considered a significant hit if its FDR was <0.25 (Supplementary Table 12).

We calculated an empirical FDR considering false positive (FP) hits as the sum of the significant lncRNA hits in the control screenings without CasRx and total hits (true positive hits (TP) + false positive hits (FP)) as the sum of the significant lncRNA hits in the correspondent condition screenings. Empirical FDR = FP/(TP + FP). For each significant hit, we calculated a gene-unified LFC. We annotated manually the gene IDs of all the hits. We grouped together lncRNA genes that corresponded to the same previously annotated gene. The gene-unified LFC value was calculated as the average of the two most significant gRNAs per gene ID (Supplementary Table 13).

## Reanalysis of previous lncRNA CRISPR screening approaches

The raw counts for the CRISPRi-based screens in all the reported cancer cell lines (HeLa, K-562, MCF7, MDA-MB-231 and U-87) were retrieved directly from the original publication[27]. An FDR for the TSS of each lncRNA was calculated using the RRA algorithm as described above. Each lncRNA FDR was defined as the most significant FDR between all the targeted TSS for each gene. A lncRNA was considered a significant hit if its FDR was <0.25. Next, we converted the TSS coordinates in the paper from Genome build hg19 to hg38 using liftover from University of California Santa Cruz[83]. The distance between each lncRNA TSS and the nearest essential coding gene or the nearest coding gene was calculated using the findOverlaps function of the GenomicRanges package[105].

A lncRNA was considered an off-target if its TSS was <1 kb from the TSS of an essential coding gene (Supplementary Table 14).

In the case of the Cas9-based deletion screens[15], we downloaded the FASTQ reads from the SRA database[106] for the Huh7.5 cell line screen (accession number SRX2148759) and for the HeLa cell line screen (accession number SRX2149095). Read counts were obtained using the MAGeCK v.0.5.9.4 count function[57] (Supplementary Table 15). FDR for each lncRNA was obtained as described above for the CasRx screens (Supplementary Table 16).

For the Cas9-based screens targeting splicing sites[16], we retrieved the raw counts from the original publication. For each gRNA, we calculated an LFC on CPM-normalized counts using the plasmid time 0 as baseline (Supplementary Table 17). To quantify the expression of the lncRNAs, we downloaded the RNA-seq FASTQ data for the cell lines HeLa and K-562 from the SRA database[106] (accession numbers SRR8615629 and SRR8615717, respectively). We obtained raw counts with Salmon v.1.5 (ref. 92) with the Gencode v.20 transcript sequences as reference. A lncRNA was defined as expressed if its TPM value was ≥0.01 (Supplementary Table 18).

## Validation of lncRNA dependencies

NT (control) gRNAs or new gRNAs against different lncRNAs were cloned into either the pLentiRNAGuide_001-hU6-RfxCas13d-DR1-BsmBI-EFS-Puro-WPRE vector (Addgene, cat. no. 138150 (ref. 83)) or the pLKO5-CasRx(DR130)-EFS-EGFP vector (Supplementary Table 10) using golden gate cloning. Cells were transduced with the vectors carrying the puromycin resistance cassette and encoding either NT gRNA or gRNAs against the lncRNAs, followed by selection with puromycin for 2 days before experiments. In the case of vectors carrying EGFP, cells were transduced at approximately 0.25 MOI and allowed to recover for 2 days with fresh medium before experiments.

For the FACS-based competition assay, after recovery, the percentage of GFP-positive cells (cells carrying a gRNA against a particular lncRNA or the NT gRNA) or untransduced cells were recorded by flow cytometry (day 0). The same cells were maintain in culture for 2 weeks and at the end of the experiment the distribution of GFP-positive cells or untransduced cells were recorded again (day 14). The experimental LFC was calculated as the fold between the proportions of GFP-positive cells at day 14 compared with day 0. Cells populations were recorded using a Beckman Coulter Cytoflex LX system. FlowJo v.10.7.2 was used to analyze populations.

For the colony formation assay, following antibiotic selection, 4,000 cells per condition were seeded into six wells, with two technical replicates per condition. After 10 days in culture, the cells were stained with crystal violet. To quantify the results, the crystal violet was dissolved in 10% acetic acid and the absorbance was measured using a GloMax Discover microplate reader (Promega).

For the proliferation assay, following antibiotic selection, 2,000 cells per condition were seeded into 96 wells, with four technical replicates per condition. For each timepoint, a separate 96-well plate was seeded using the same cell master mix for all conditions. Surviving cells were quantified based on luminescence using CellTiter-Glo, after 8 h (day 0), 1 day, 2 days or 3 days of culture, using a GloMax Discover microplate reader (Promega). The fluorescence values representing the number of cells were normalized to day 0 and are indicated as fold change over the controls (cells transduced with a NT gRNA).

## Data availability

All sequencing datasets have been deposited in ENA (accession number: PRJEB60776). The GENCODE Human Release 38 genome (GRCh38.p13) and transcriptome can be found here: https://www.gencodegenes.org/human/release_38.html. The resources from DepMap and the CCLE can be found here: https://doi.org/10.6084/m9.figshare.19700056.v2. The resources from RNAcentral can be found here: https://ftp.ebi.ac.uk/pub/databases/RNAcentral/releases/16.0.

## Code availability

The custom pipelines for CasRx gRNA arrays design, gRNA arrays quantification from amplicon-seq data and the computational analysis of the CasRx lncRNAs screens used in this paper are available at Github: https://github.com/roland-rad-lab/Genome-scale-pan-cancer-interrogation-of-lncRNA-dependencies-using-CasRx.git.

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

## Acknowledgements

We thank J. Eichinger, A. Grotloh and V. Aigner for excellent technical assistance. This study was supported by the European Research Council (Consolidator grant CoG PACA-MET-819642 and MSCA-ITN-ETN-861196 to R.R.; CoG no. 648521 to D.S.); the Deutsche Forschungsgemeinschaft (DFG RA1629/2-1;SFB1321 to R.R. and M.S.S.; and SFB 1371 to D.S.); German Cancer Consortium, Deutsche Krebshilfe (70114314 to R.R.); the German Federal Ministry of Education and Research (Cluster4Future: CNATM to R.R.) and TUM Innovation Network NextGenDrugs funded under the Excellence Strategy of the Federal Government and the Länder (to R.R.). J.J.M. was supported by a European Molecular Biology Organization (EMBO) long-term fellowship (ALFT 655-2019).

## Author contributions

J.J.M. and R.R. designed the study. J.J.M., R.T., M.S., R.Ö., A.B., E.Z. and P.W. carried out the research. J.J.M. and R.T. analyzed the data. R.T. developed the code. R.T. and T.E. performed the bioinformatic analyses. R.R., M.S.S. and D.S. supplied resources. J.J.M. and R.R. supervised the project. J.J.M. and R.T. prepared the figures. J.J.M. and R.R. wrote the manuscript.

## Funding

## Competing interests

The authors declare no competing interests.

## Additional information

**Correspondence and requests for materials** should be addressed to Juan J. Montero or Roland Rad.

| | |
|---|---|

# Reporting Summary

## Statistics

For all statistical analyses, confirm that the following items are present in the figure legend, table legend, main text, or Methods section.

| n/a | Confirmed | |
|---|---|---|
| ☐ | ☒ | The exact sample size (*n*) for each experimental group/condition, given as a discrete number and unit of measurement |
| ☐ | ☒ | A statement on whether measurements were taken from distinct samples or whether the same sample was measured repeatedly |
| ☐ | ☒ | The statistical test(s) used AND whether they are one- or two-sided *Only common tests should be described solely by name; describe more complex techniques in the Methods section.* |
| ☐ | ☒ | A description of all covariates tested |
| ☐ | ☒ | A description of any assumptions or corrections, such as tests of normality and adjustment for multiple comparisons |
| ☐ | ☒ | A full description of the statistical parameters including central tendency (e.g. means) or other basic estimates (e.g. regression coefficient) AND variation (e.g. standard deviation) or associated estimates of uncertainty (e.g. confidence intervals) |
| ☐ | ☒ | For null hypothesis testing, the test statistic (e.g. $F$, $t$, $r$) with confidence intervals, effect sizes, degrees of freedom and $P$ value noted *Give P values as exact values whenever suitable.* |
| ☒ | ☐ | For Bayesian analysis, information on the choice of priors and Markov chain Monte Carlo settings |
| ☐ | ☒ | For hierarchical and complex designs, identification of the appropriate level for tests and full reporting of outcomes |
| ☐ | ☒ | Estimates of effect sizes (e.g. Cohen's *d*, Pearson's *r*), indicating how they were calculated |

*Our web collection on statistics for biologists contains articles on many of the points above.*

## Software and code

Policy information about availability of computer code

| Data collection | No software was used |
|---|---|
| Data analysis | featureCounts (subread package v1.6.3), Salmon v1.5.2, limma v3.42.2, MAGeCK v0.5.9, https://gitlab.com/sanjanalab/cas13 |

For manuscripts utilizing custom algorithms or software that are central to the research but not yet described in published literature, software must be made available to editors and reviewers. We strongly encourage code deposition in a community repository (e.g. GitHub). See the Nature Portfolio guidelines for submitting code & software for further information.

## Data

Policy information about availability of data

All manuscripts must include a data availability statement. This statement should provide the following information, where applicable:
- Accession codes, unique identifiers, or web links for publicly available datasets
- A description of any restrictions on data availability
- For clinical datasets or third party data, please ensure that the statement adheres to our policy

Data Availability
All sequencing datasets have been deposited in ENA. ENA accession number: PRJEB60776 (https://www.ebi.ac.uk/ena/browser/view/PRJEB60776).
The GENCODE Human Release 38 genome (GRCh38.p13) and transcriptome can be found here:
https://www.gencodegenes.org/human/release_38.html.

## Human research participants

Policy information about studies involving human research participants and Sex and Gender in Research.

| | |
|---|---|
| Reporting on sex and gender | n/a |
| Population characteristics | n/a |
| Recruitment | n/a |
| Ethics oversight | n/a |

Note that full information on the approval of the study protocol must also be provided in the manuscript.

# Field-specific reporting

Please select the one below that is the best fit for your research. If you are not sure, read the appropriate sections before making your selection.

☒ Life sciences        ☐ Behavioural & social sciences        ☐ Ecological, evolutionary & environmental sciences

For a reference copy of the document with all sections, see nature.com/documents/nr-reporting-summary-flat.pdf

# Life sciences study design

All studies must disclose on these points even when the disclosure is negative.

| | |
|---|---|
| Sample size | The sample size is reported in the figure legends for each individual experiment. We have no any kind of experiments (as animals base ones) that requiered sample size calculations. Most of our experiments are screenings, with intenal controls and independent gRNAs that validate the experiment statistically. For other kind of experiments (lncRNA validations), we validate each lncRNA with several different methods (colony formation, proliferation curves and RNA-seq). All the experiments behave in the same way, making the need of extra sample size innecesary. |
| Data exclusions | No data was excluded. |
| Replication | All attempt of replication were successful. How many times each experiment was replicates is reported in the figure legends of each individual experiment. |
| Randomization | Most of our experiments included internal controls, making randomizing the samples unnecessary. The final outputs are independent of the researcher, as NGS is inherently unbiased. |
| Blinding | Most of our experiments included internal controls, making blinding unnecessary. The final outputs are independent of the researcher, as NGS is inherently unbiased. |

# Reporting for specific materials, systems and methods

We require information from authors about some types of materials, experimental systems and methods used in many studies. Here, indicate whether each material, system or method listed is relevant to your study. If you are not sure if a list item applies to your research, read the appropriate section before selecting a response.

## Materials & experimental systems

| n/a | Involved in the study |
|---|---|
| ☒ | ☐ Antibodies |
| ☐ | ☒ Eukaryotic cell lines |
| ☒ | ☐ Palaeontology and archaeology |
| ☒ | ☐ Animals and other organisms |
| ☒ | ☐ Clinical data |
| ☒ | ☐ Dual use research of concern |

## Methods

| n/a | Involved in the study |
|---|---|
| ☒ | ☐ ChIP-seq |
| ☐ | ☒ Flow cytometry |
| ☒ | ☐ MRI-based neuroimaging |

# Eukaryotic cell lines

Policy information about cell lines and Sex and Gender in Research

| | |
|---|---|
| Cell line source(s) | HEK-293T, KP-4 and MIA PaCa-2 were available in our research center. A549 and NCI-460 were provided by Mariano Barbacid from the Spanish National Cancer Research Center (CNIO). LN18 and LN-229 were provided by massimo Squatrito from the Spanish National Cancer Research Center (CNIO). |
| Authentication | None of the cell lines were authenticated. |
| Mycoplasma contamination | All the cell lines were tested regularly for mycoplasma contamination. And all of them were negative. |
| Commonly misidentified lines (See ICLAC register) | None of the cell lines were authenticated. |

# Flow Cytometry

## Plots

Confirm that:

☒ The axis labels state the marker and fluorochrome used (e.g. CD4-FITC).

☒ The axis scales are clearly visible. Include numbers along axes only for bottom left plot of group (a 'group' is an analysis of identical markers).

☒ All plots are contour plots with outliers or pseudocolor plots.

☒ A numerical value for number of cells or percentage (with statistics) is provided.

## Methodology

| | |
|---|---|
| Sample preparation | Cells were resuspended in FACS Buffer (PBS, 3% FBS, 5mM EDTA). |
| Instrument | Beckman Coulter Cytoflex LX system. |
| Software | FlowJo v10.7.2 |
| Cell population abundance | A minimum of 10000 cells were recorded. |
| Gating strategy | Singles were selected using FSC-width and FSC-A values, tRFP657 cells were selectd using SSC-A and Y675-PC5-A values, finally GFP intensity in the tRFP657 positives cells were measured as the mean intensity of the B525-FITC-A values. |

☒ Tick this box to confirm that a figure exemplifying the gating strategy is provided in the Supplementary Information.

