## [Peer Review File · Nature Methods]

Peer Review Information

Manuscript Title: Genome-scale pan-cancer interrogation of lncRNA dependencies using CasRx

Corresponding author name(s): Roland Rad, Juan J. Montero

Editorial Notes: None

Reviewer Comments & Decisions:

Decision Letter, initial version:

Dear Ronald,

Your Article, "Genome-scale pan-cancer interrogation of lncRNA dependencies using CasRx", has now been seen by 3 reviewers. As you will see from their comments below, although the reviewers find your work of considerable potential interest, they have raised a number of concerns.

We are interested in the possibility of publishing your paper in Nature Methods, but would like to consider your response to these concerns before we reach a final decision on publication. We therefore invite you to revise your manuscript to fully address these concerns.

* include a point-by-point response to the reviewers and to any editorial suggestions

* please underline/highlight any additions to the text or areas with other significant changes to facilitate review of the revised manuscript

* address the points listed described below to conform to our open science requirements

* ensure it complies with our general format requirements as set out in our guide to authors at www.nature.com/naturemethods

* resubmit all the necessary files electronically by using the link below to access your home page

[Redacted] This URL links to your confidential home page and associated information about manuscripts you may have submitted, or that you are reviewing for us. If you wish to forward this email to co-authors, please delete the link to your homepage.

We hope to receive your revised paper within 8 weeks. If you cannot send it within this time, please let us know. In this event, we will still be happy to reconsider your paper at a later date so long as nothing similar has been accepted for publication at Nature Methods or published elsewhere.

OPEN SCIENCE REQUIREMENTS

REPORTING SUMMARY AND EDITORIAL POLICY CHECKLISTS

DATA AVAILABILITY

All novel DNA and RNA sequencing data, protein sequences, genetic polymorphisms, linked genotype and phenotype data, gene expression data, macromolecular structures, and proteomics data must be deposited in a publicly accessible database, and accession codes and associated hyperlinks must be provided in the “Data Availability” section.

Please include a “Data availability” subsection in the Online Methods. This section should inform readers about the availability of the data used to support the conclusions of your study, including accession codes to public repositories, references to source data that may be published alongside the paper, unique identifiers such as URLs to data repository entries, or data set DOIs, and any other statement about data availability. At a minimum, you should include the following statement: “The data that support the findings of this study are available from the corresponding author upon request”, describing which data is available upon request and mentioning any restrictions on availability. If DOIs are provided, please include these in the Reference list (authors, title, publisher (repository name), identifier, year). For more guidance on how to write this section please see: <http://www.nature.com/authors/policies/data/data-availability-statements-data-citations.pdf>

MATERIALS AVAILABILITY

SUPPLEMENTARY PROTOCOL

To help facilitate reproducibility and uptake of your method, we ask you to prepare a step-by-step Supplementary Protocol for the method described in this paper. We [encourage authors to share their step-by-step experimental protocols](https://www.nature.com/nature-research/editorial-policies/reporting-standards#protocols) on a protocol sharing platform of their choice and report the protocol DOI in the reference list. Nature Portfolio 's Protocol Exchange is a free-to-use and open resource for protocols; protocols deposited in Protocol Exchange are citable and can be linked from the published article. More details can found at www.nature.com/protocolexchange/about.

ORCID

Sincerely,
Madhura

Madhura Mukhopadhyay, PhD
Senior Editor
Nature Methods

Reviewers' Comments:

Reviewer #1:

Remarks to the Author:

The authors describe the development of a large-scale screening system for lncRNA loss-of-function in cancer cells, based on the CasRx-mediated knockdown, that, if successful, can overcome the clear limitations of the other available systems, which include those based on Cas9 (limited efficiency), CRISPRi (problems with adjacent promoters/enhancers) shRNAs (limited nuclear activity) or GapmeRs (can't be pooled). The authors describe a vector system that allows strong CasRx expression and apparently strong on-target activity, a very large target group of lncRNAs (potentially too large, see below), and results from applying a large library in the context of cancer cells with some validation of few hits.

Overall, the presented work is impressive. The library and/or the expression system will potentially be useful to many researchers interested in lncRNAs, both in the context of cancer and otherwise. However, more data should be presented to make a convincing case that the system indeed works as advertised. In particular, off-target effects need to be evaluated further. The authors present some data supporting limited off-targeting, but more analyses/experiments can make the data stronger:

A) The authors choose to go with a very comprehensive approach for including lncRNAs in their libraries. While lncRNA expression levels go into consideration, the thresholds for eventual selection appear to be very low, as the authors indicate that as many as 10,000 lncRNAs from the library are expressed in each of the CCLE cell lines, whereas using traditional thresholds of ~1 copy per cell, this number is typically ~1,000. The threshold of coverage of 90% of the exon by 1 read across *all* CCLE lines is very low. Then the authors admit multi-mapping reads when quantifying expression, and since most lncRNAs extensively overlap transposable elements, presumably much of the expression is coming from these regions. Lastly, a TPM threshold of 0.01 is ridiculously low. 1 TPM corresponds approximately to 1 copy per cell (within an order of magnitude). What is the biological meaning of a transcript expressed (in a cell line that is at least somewhat uniform) at 0.01 TPM? This should be at least discussed. Also, it should be explained how the authors deal with transposable elements. More importantly, are the cell-type specific positive hits in the screen indeed expressed at reasonable levels in the cell lines where they

were discovered more than in the other cell lines? This should be an important and easy test to validate the specificity of the approach and its ability to recover biologically relevant lncRNA targets. Relatedly – are the never-essential genes targeted by the NE gRNAs indeed expressed in the cells the authors are working with?

B) What is the level of concordance between different gRNAs targeting the same lncRNA? The authors present the use of limited number of gRNAs per gene as an advantage because it allows for smaller libraries, but there is also a shortcoming in that the confidence in individual hits (and in particular due to potential off-targets) is lower.

C) The current validation is based on few genes, mostly well-expressed lncRNAs with orthogonal support for their functions. A secondary library, containing other gRNAs targeting the dozens of other positive hits in the screen using CasRx alongside additional shuffled controls can potentially further strengthen the results.

D) How many of the initial hits of the screen were removed by the “blacklisting” based on potential off-targets? If the authors use shorter kmers of 22 or 21 nts, are there additional suspected off-targets? Are these potential off-targets enriched with gRNAs where consistency between the different gRNAs targeting the same lncRNA is lower (i.e., is their correlation between confidence in no-off-targets and confidence derived from similarity between gRNAs? These are key aspects to address the potential off-targets that are a major concern when, as the authors indicate, the rate of positive hits is lower than in traditional screens targeting protein coding genes, and in addition, many of the lncRNAs the authors are targeting are apparently expressed at extremely low levels in the cells they work with.

E) In Figure 4a, the authors should show the distributions of the lncRNAs alongside the distributions of the positive and negative controls. Is there any difference between lncRNAs as a group and the negative controls?

Minor comments:

A) Line 63: “because cis effects are splicing-independent” – it's not clear what is meant here, as lncRNA splicing was shown by several studies to be relevant to the activity of cis-acting lncRNAs.

B) Line 66: “lower numbers of true-positive dependencies than” – it is not clear enough what the authors mean by “dependencies” here. “Cellular phenotypes” would be better.

C) Line 149: “genomic occupancy, sequence similarity, and transcript directionality” – this description is not clear, as the authors don't really use sequence similarity, overlap in genomic co-ordinates. It is also unclear why they call these “lncRNA families” rather than “lncRNA genes”, as every family is essentially a gene with different transcript isoforms arising from it, even if the gene groups are not exactly the same as in GENCODE because of the slightly different grouping criteria.

D) Line 222: the authors should explain that the barcode is used for sample identification and multiplexing, its not clear from the text.

Reviewer #2:

Remarks to the Author:

The manuscript by Montero et al. describes a high throughput strategy to leverage genome integrated CRISPR/CasRx for genome-wide depletion of lncRNAs in various cancers. They used transposons to insert multiple copies of CasRx into cancer genome with a gRNA library targeting 24,171 lncRNA families and 158,816 transcripts. Smaller scale of noncoding RNA screening has been demonstrated before and the main novelty of this work is the genome-wide scale of the screening. The study is overall interesting but a few concerns must be addressed.

Major:

(1) Using transposons to deliver CasRx is highly efficient. However, the landing sites could be random which may lead to high variations in the screening. For example, the insertion could occur at different loci in different replicative experiments. Therefore, the screening results may not be reproducible due to synthetic impact by disrupted genes in adjacent to the insertions. This might be already reflected in their own analysis in Supplementary Figure 3b. i.e. Pearson correlation between replicative experiments is very low (equivalent to that between different cancer types, $\sim 0.4-0.5$). This type of variation is irrelevant to the expression level of CasRx gene but more likely by different genes affected by the insertion. The cells from replicative experiments can behave very differently even if the expression of CasRx is similar. The authors should carefully address the issue of variation and low reproducibility caused by random insertion. As a methodology paper, they should give a clear instruction on how to avoid such problems.

(2) The design of gRNA is not clear. For example, (a) how the pair of gRNAs targeting the same transcript is chosen? (b) The number of gRNA pairs only depends on the number of exons, instead of the length and secondary structure of the transcript, which I feel are more important features to be considered. They should perform the screening in the same cells with an independent set of gRNAs and compare the results. Further analysis of the difference between the two independent screenings may provide valuable information on the impact of gRNA coverage/design.

(3) Delivery of gRNA to cell lines is always a critical parameter in screening work. The authors failed to show the coverage of gRNAs in each screen experiment as a very essential quality control. Not every lncRNA could be screened if the coverage in some cell line is low. This may also explain the low Pearson correlation in Supplementary Fig. 3b. Furthermore, transduced cells before antibiotic selection were collected as control samples (e.g. Liu, Horlbeck et al. 2017, Liu, Cao et al. 2018). It is rational to involve such control to calculate the LFC.

(4) The inclusion of lncRNAs in the screening mostly depends on their expression level in various cancer cells. Even though they show in figure 3 the enrichment of functional traits in the targeted lncRNAs, it remain an issue that some modestly expressed essential lncRNAs are missing from the screening. What would be the limiting factor for the capacity of the screening? Why not include all? They should at least

do saturation screening in one cancer cell type and compare the results with the Albarossa library to show if any essential one is missing.

(5) CasRx is a class 2 type VI CRISPR-Cas RNA endonuclease. That means the targeted RNA is not necessarily degraded by the enzyme lacking exonuclease activity. The RNA may be simply cut into two fragments. This is not a problem when screening coding gene as cutting off the start codon is adequate to suppress the mRNA translation. However, the targeted noncoding RNA may still be partially or fully functional, causing high ratio of false negative. Comparison of the screening result with the ASO-based screening will give us clear clue about this concern.

(6) Due to the inherent problem of CasRx discussed above, using the same enzyme (CasRx) for validation may not be appropriate (Fig. 5). The authors may want to use orthogonal methods for validation of their findings. For example, ASO-based, RNAi-based and CRISPRi-based knockdown could be considered. In addition, rescue experiment can be performed to confirm that the phenotype was indeed caused by lncRNA deficiency.

(7) Proliferation is only one characteristics of cancer. The lncRNAs screened by the current design may not be cancer specific. Authors should consider using some normal cells as background controls, and they may identify some essential genes for general cell proliferation (which is less interesting to this study purpose). Furthermore, the authors could consider screening metastasis, or other cancer hallmark feature related lncRNAs.

(8) The potential impact of the genome-integrated CasRx protein in human cancer cell lines. However, it has been reported that CasRx protein has collateral activity, which is detrimental to cell proliferation (Shi, Murphy et al. 2023). It seems rational to apply an inducible system to control the CasRx protein expression to prevent the side effects from the constitutive expression of CasRx proteins. In addition, the authors should carry out RNA-seq assay in those transgenic and parental cell lines with or without target gRNA transduction to figure out differential genes may be involved in function lncRNA screening.

Minor:

(9) Figure 5g, is the statistics (mean, SD and P values) from repeated quantification or independent experiments?

(10) In Data Availability, hyperlink to the data reservoir should be provided.

(11) The numeric value of y-axis in Supplementary Figure 4f should be 150, not "1.50".

(12) Line 545, should be HEK-293T, not "HEK-2937T".

Reviewer #3:

Remarks to the Author:

The manuscript by Montero et al. reports a novel Cas13/CasRx-based screen to systematically and unbiasedly identify lncRNAs with potentially important functions in multiple cancer cell lines. The authors developed a genome-integrated CasRx system using different cancer lines to achieve genome-scale knock-down of lncRNAs. As depletion of lncRNA transcripts is a challenging task, the study presents an elegant system to screen for lncRNA loss of functions, overcoming limitations of several other studies. I believe it is an important and well-designed study; the results of the screen will be of the interest for the lncRNA community for further follow-up analyses. The manuscript is very well-written. My specific comments to clarify some aspects of the study and improve its readability are summarized below.

Major comments on data presentation:

1. I found that there are too many technical details in the results describing library design, generation, sequencing and mapping lncRNA dependencies. These sections can be shortened and repetitive technical details can be moved to the material and methods. Shortening these parts would improve the readability and accessibility of the manuscript.
2. The authors should elaborate on the fitness screening and give details of their assays and readouts. Currently, the authors only cite publications that used the same or similar approaches.
3. The functional data presented in Figure 5 should be expanded and explained in the results section. Description of cellular assays used for functional assessment and validation of screen hits is missing. They should be mentioned in results and included in the method section.
4. In my opinion, what is missing is a figure illustrating the overview of the screen. The experimental design of the screen gets lost in technical details of individual steps. I would recommend to introduce a nice scheme including a summary of cell lines which were used, the constructs integrated and/or transfected to perform the actual screen, what read-outs were used etc.

Further specific comments:

5. Optimization of a genome-integrated CasRx system

5.1 The authors should list the CasRx sequence in supplemental files and also clarify if the NLS was added to the sequence.

5.2 From the description of the genome-integrated CasRx system, it was not clear if the system would work for both, nuclear and cytoplasmic transcripts.

5.3 The authors should add small-scale control experiments showing that their CasRx system is efficient for depletion of nuclear (ie. Malat1) and cytoplasmic lncRNAs. A few abundant transcripts would be sufficient.

6. Validation of lncRNA dependencies

6.1 The authors should mention what is the proportion of nuclear vs. cytoplasmic lncRNA transcripts which were identified in the screen (Figure 5d). Is there a bias of the screen results towards nuclear enriched transcripts?

6.2 For the selected six lncRNAs, the authors should demonstrate changes in their expression levels upon CasRx-depletion by a simple qRT-PCR or similar methods.

Author Rebuttal to Initial comments

Point by point responses to reviewer's comments. Montero, Trozzo et al, "**Genome-scale pan-cancer interrogation of lncRNA dependencies using CasRx**".

We would like to thank the reviewers for their thoughtful comments on the manuscript, which have helped to improve the quality of the work. We endorsed the suggestions and performed a large series of new experiments and analyses.

The table below gives an overview of new figure additions to the revised manuscript.

New Data	Reviewer comment	Content
Fig. 1a	Rev3, C5.1, C5.2	Schematic representation of the episomal HypBase transposase and the transposon vectors used to deliver CasRx. Now the scheme shows the NLS fused to the CasRx protein.
Fig. 1c, Supplementary Fig. 1a	Rev2, C8	Full-length deep RNA-Seq analysis of the parental cell culture and related CasRx transgenic cell clones. The data prove that CasRx transgenic cell clones are highly similar to the parental cell lines.
Supplementary Fig. 1b	Rev2, C1	Agarose gel electrophoresis displaying the PCR amplification of the HypBase-transposase or the CasRx-transposon locus using either the HypBase-transposase vector or DNA extracted from the CasRx clones as template.
Supplementary Fig. 1c	Rev2, C1	Quantitative transposon insertion site sequencing (QiSeq) used to identify the number and genomic localization of the CasRx-transposon insertions.

Fig. 1h,i	Rev1, CA Rev2, C8	RNA-seq analysis of different knockdown experiments targeting GFP or endogenous genes. The data prove the lack of global gene dysregulation upon CasRx on-target cleavage.
Fig. 1j, Supplementary Fig. 1d,e	Rev1, CA Rev2, C8	Analysis of 3 screens performed using a newly generated CasRx control library. The library have been designed mining DepMap data and is composed of gRNAs targeting never-essential expressed coding genes and gRNAs targeting always-essential coding genes.
Fig. 2e	Rev1, CB	gRNA sequence coverage (percentage of lncRNA sequence targeted by its specific gRNAs) for the lncRNA families targeted by the Albarossa library.
Supplementary Fig. 3a	Rev1, CA	Comparison between the two different methods used to quantify lncRNA families expression.
Supplementary Fig. 3b	Rev1, CA	Median TPM distribution for both lncRNAs and coding genes across CCLE cell lines. This information was used to select a universal expression threshold for the lncRNA families.
Supplementary Fig. 3c	Rev1, CA	Average number of lncRNAs expressed at different thresholds (>0.1 and >1 TPM) across tissues.
Fig. 4a	Rev3, C2, C4	Schematic representation of the experimental outline of CasRx-based genome-scale lncRNA fitness screening developed in this study
Supplementary Fig. 4b	Rev2, C3	Distribution of the gRNAs counts in each screen.
Supplementary Fig. 4d	Rev2, C1, C3	Re-analysis of the Pearson correlation between replicates of previously reported lncRNA screens (Zhu et al, Nature biotechnology 2016. Liu et al, Science 2017. Liu et al, Nature biotechnology 2018).

Supplementary Fig. 4e	Rev2, C1, C3	Analysis of the Pearson correlations between screen replicates of genes that are known to have biological effects or genes without biological relevance.
Fig. 4d, Supplementary Fig. 5e	Rev1, CA, CE Rev2, C8	LFC distribution of negative control gRNAs and gRNAs targeting expressed or non-expressed lncRNA families.
Fig. 4g, Supplementary Fig. 6d,e	Rev2, C4	Analysis of a screen performed using a newly generated CasRx LN-18 Δ Saturation library. The Δ Saturation library targets all lncRNA families expressed in the LN-18 cell line (TPM >0.01) that are not targeted by the Albarossa library (to achieve saturation mutagenesis).
Fig. 4h	Rev2, C4	% of discovery (number of hits among expressed lncRNA families) for the Albarossa or the LN-18 Δ Saturation libraries.
Supplementary Fig. 6c	Rev1, CB	Percentage of gRNA arrays contributing to the phenotype (fitness defect) of lncRNA and protein-coding gene screen hits.
Fig. 5a	Rev3, C5.2, 5.3, 6.1	Full-length deep RNA-Seq analysis of the nuclear and cytoplasmic fractions for all the CasRx transgenic cell clones.
Fig. 5f	Rev1, CA	Expression analysis of the screen hit or non-hit lncRNA families per cell line.
Fig. 5g	Rev1, CA	Expression analysis of cell line specific hits in the cell line in which they are a hit or in the other cell lines.
Fig. 6a, Supplementary Fig. 6e.	Rev1, CC Rev2, C2b Rev3, C3	Screening-based high-throughput validation of lncRNA dependencies using a newly generated "validation library" that targets 93 hits from the previous screens.

Fig. 6b	Rev1, CC Rev2, C2b Rev3, C3, C5.3	Individual hit validation using FACS-based competition assays in different cell lines. We validated 16 lncRNA families including nuclear and cytoplasmic ones.
Fig. 6e	Rev3, C6.2	qPCR-based quantification of CasRx knock-down levels for different lncRNAs

Reviewers' Comments:**Reviewer #1:****Remarks to the Author:**

The authors describe the development of a large-scale screening system for lncRNA loss-of-function in cancer cells, based on the CasRx-mediated knockdown, that, if successful, can overcome the clear limitations of the other available systems, which include those based on Cas9 (limited efficiency), CRISPRi (problems with adjacent promoters/enhancers) shRNAs (limited nuclear activity) or GapmeRs (can't be pooled). The authors describe a vector system that allows strong CasRx expression and apparently strong on-target activity, a very large target group of lncRNAs (potentially too large, see below), and results from applying a large library in the context of cancer cells with some validation of few hit.

Overall, the presented work is impressive. The library and/or the expression system will potentially be useful to many researchers interested in lncRNAs, both in the context of cancer and otherwise. However, more data should be presented to make a convincing case that the system indeed works as advertised. In particular, off-target effects need to be evaluated further. The authors present some data supporting limited off-targeting, but more analyses/experiments can make the data stronger:

A) The authors choose to go with a very comprehensive approach for including lncRNAs in their libraries. While lncRNA expression levels go into consideration, the thresholds for eventual selection appear to be very low, as the authors indicate that as many as 10,000 lncRNAs from the library are expressed in each of the CCLE cell lines, whereas using traditional thresholds of ~1 copy per cell, this number is typically ~1,000. The threshold of coverage of 90% of the exon by 1 read across *all* CCLE lines is very low. Then the authors admit multi-mapping reads when quantifying expression, and since most lncRNAs extensively overlap transposable elements, presumably much of the expression is coming from these regions. Lastly, a TPM threshold of 0.01 is ridiculously low. 1 TPM corresponds approximately to 1 copy per cell (within an order of magnitude). What is the biological meaning of a transcript expressed (in a cell line that is at least

somewhat uniform) at 0.01 TPM? This should be at least discussed. Also, it should be explained how the authors deal with transposable elements.

The reviewer raises important questions on methodological aspects, which we will individually answer in detail below.

1. Overlap with repetitive and transposable elements.

We agree that repetitive elements can be an important confounder when quantifying lncRNA expression. In fact, in our initial analyses, we already explicitly addressed this concern, but we probably failed to explain adequately our methodological approach in the initial version of the manuscript. In brief:

We utilized two distinct methods to analyse lncRNA expression:

(a) A first “sensitive/permissive” approach (STAR in combination with featureCounts) served as the basis for selection of lncRNA families for CRISPR library design. The key goal at this stage was to not “lose” overlapping lncRNAs, for which accurate quantification is very difficult (and which are prone to be filtered out by more stringent approaches). In this approach we first mapped reads to the genome using STAR v2.7.5b. Subsequently, the mapped reads were fed to featureCounts to quantify lncRNA expression. We allowed for multi-overlap of features, ensuring that if a read aligns to a location encoding multiple features (overlap of more than one lncRNA family), it is counted once for each feature. In contrast to protein-coding genes, this scenario is very common for lncRNAs. In this step, we excluded multi-mapping reads, thereby avoiding inclusion of reads aligning to multiple genomic positions, such as transposable elements.

(b) A second method for more “stringent” quantification of lncRNA expression served as a basis for a “fair” calculation of the number of expressed lncRNA per cell line. While featureCounts proved very useful for lncRNA selection to be considered for library design, we were aware of its potential to overestimate the number of expressed lncRNAs per cell line. Therefore, to be conservative in determining the number of Albarossa-targeted lncRNAs that are expressed in each cell line, we used Salmon. Salmon employs a sophisticated statistical approach to quantify overlapping regions (Patro et al., Nature methods, 2017), ensuring that expression is not overestimated (individual reads are not being double-counted). To prevent spurious read mapping, particularly from repetitive regions, Salmon supports the use of the full genome as a decoy. Using this functionality, we ensured that reads aligning better to the genome than to the transcriptome are discarded, thereby preventing false-positive detection of lncRNAs annotated into repetitive regions.

To validate the explained behavior of both methods, we show the median expression of each lncRNA family across CCLE cell lines, as determined by featureCounts versus Salmon (Supplementary Fig. 3a). The correlation is very high but – as expected – a small subset (5.9%) of lncRNAs is overestimated by featureCounts. These lncRNA genes exhibit substantial overlap with other lncRNAs (overlap indicated with red colour). Thus, featureCounts allowed us to achieve our first goal for library design (not “losing” overlapping lncRNAs, that are reasonably expressed, but out-filtered by other approaches). In contrast, Salmon shows minimal lncRNA overestimation, demonstrating the functionality of the “decoy approach” in removing spurious reads originating from repetitive elements.

The above is now explained in the results section (page 5, line 212), in more detail the methods section (page 15, lines 712-739), and in the supplementary figure legends (Supplementary Fig. 3a). The data are presented in Supplementary Fig. 3a.

2. Filtering of non-expressed exons.

Regarding the reviewer’s comment “The threshold of coverage of 90% of the exon by 1 read across *all* CCLE lines is very low”: it is important to clarify that this criterion is merely a pre-annotation filtering step to remove possible “artefactual” exons. Please note that the selection of lncRNA families for our library design relies on different downstream criteria, which will be discussed below (point 3).

Nevertheless, we would like to explain here why we deployed this filtering step. We used very comprehensive datasets (from RNA central and other sources) that include multiple isoforms per lncRNA locus, some of which have low abundance and might originate from RNA-processing errors such as intron-retention. We intended to remove such “artefacts”. Since introns are generally longer than exons and the calculation of transcripts per million (TPM) considers gene length, we used the criterium pointed out by the reviewer to exclude non-expressed exons from our annotation and ensure a fair calculation of each lncRNA family's expression. This has been now clarified in the methods section (page 14, lines 685-689).

As a side note, we would also like to point out that during later library design, we used an additional approach to avoid targeting of low-abundance isoforms that might not have been filtered out during the initial filtering step. This is outlined in the methods section (page 16, lines 782-787) as follows: “To maximize the targeting of the most expressed isoforms we avoided designing gRNAs against low-expressed exons. We quantified exon expression levels (normalized by length) using featureCounts (subread package v1.6.3) with the -f option to output exon-level instead of transcript-level counts (Supplementary Table 8). We

considered an exon as low-expressed if its normalized expression level was less than two standard deviations from the mean normalized expression level of all the exons from the same lncRNA family.”

3. Selection of an expression threshold.

We acknowledge that the 0.01 TPM threshold is low. We chose this threshold because of several reasons. A common approach to select a threshold that separates expressed from non-expressed genes considers the distribution of expression values in the relevant model system. We considered this important, as our model system (cancer cell lines) displays high rates of proliferation and transcription. Therefore, whilst 1 TPM can indicate one transcript per cell in some contexts (ref), this might not be true for our system.

In order to select a universal expression threshold, we plotted the median TPM distribution for both lncRNAs and coding genes across CCLE cell lines. As shown in Supplementary Fig. 3b, 0.01 TPM effectively discriminates expressed from non-expressed lncRNAs, while 0.1 TPM serves as a moderate threshold, and 1 TPM proves overly stringent. Even for coding genes (which are by average expressed 10 times higher than lncRNAs (Derrien, et al, *Genome Res.* 2012)), the threshold of 1 TPM (cutting centrally through the distribution curve) seems too stringent in our model system. Rather, 0.1 TPM seems more adequate.

We agree with the reviewer that the most interesting lncRNAs are those highly expressed. To provide readers with a more detailed information on the expression of lncRNAs targeted by our library, we added Supplementary Fig. 3c. We now display the average number of lncRNAs expressed at different thresholds (>0.01, >0.1 and >1 TPM) across tissues. It is now clear for the reader that our library targets on average 12510 lncRNAs with expression above 0.01 TPM (Fig. 3c), 8192 lncRNAs above 0.1 TPM and 2357 lncRNAs above 1 TPM (Supplementary Fig. 3c). This information has been added to the results section (page 5, lines 217-219). The data is shown in Supplementary Fig. 3c.

Furthermore, we would like to answer to the reviewer’s comment that the number of expressed lncRNAs with more than one transcript per cell should be around 1000 (Comment: “using traditional thresholds of ~1 copy per cell, this number is typically ~1,000”). Indeed, in Genecode an average of 700 lncRNAs expressed >1 TPM can be found per tissue (Reviewer fig. 1a), closely resembling the reviewer's expectation. However, whilst the Genecode annotation is well curated, it is incomplete. We therefore used the most comprehensive annotation, encompassing 7 times as many lncRNAs as compared to Genecode. This likely explains, why we find 4.8 times as many lncRNAs expressed >1 TPM in our collection (3398 lncRNAs) (Reviewer fig. 1b) and also underscores the strength of our expression analyses.

Lastly, we would like to clarify that none of these thresholds were applied in the selection of the lncRNA families. Our selection was based on an iterative process, primarily driven by the criterion of expression. In

essence, we selected lncRNA families with the greater average expression levels across all CCLE cell lines, the greater average expression within specific tissues, and the greater expression in individual cell lines. It is worth noting that the “maximum median expression” (using the maximum expression value for each lncRNA across CCLE lines to calculate a median) for our selected lncRNA families is 5.36 TPM (Fig. 3f). This shows that lncRNAs targeted by our library exhibit robust expression in at least one cancer cell line. This consideration is important, as we want our library to be applicable across cell lines. This information is provided in the figure legends (Fig. 3f). The data is shown in Figure 3f.

Reviewer comment: More importantly, are the cell-type specific positive hits in the screen indeed expressed at reasonable levels in the cell lines where they were discovered more than in the other cell lines? This should be an important and easy test to validate the specificity of the approach and its ability to recover biologically relevant lncRNA targets.

To address this comment, we performed deep full-length RNA-seq for all cell clones used for screening. We first compared within each cell line the average expression of lncRNAs identified as hits to the average expression of “non-hit” lncRNAs. We indeed found that expression of lncRNA hits is 4.5-8.7 times higher as compared to non-hit lncRNAs (Fig. 5f).

Second, we compared the average expression of cell line-specific hits (lncRNAs that are a hit in only one cell line) to their average expression in the other cell lines. Similarly to the above “within-cell line comparisons”, we observed a strongly increased expression of lncRNAs in cell lines in which they are a hit as compared to cell lines in which they are not causing fitness defects (Fig. 5g). This supports the notion that our screens are identifying biologically relevant lncRNAs, in line with the reviewer's suggestion.

The above is now explained in the results section (page 8, lines 347-348 and lines 368-370). The data are presented in Figures 5e,g.

Reviewer comment: Relatedly – are the never-essential genes targeted by the NE gRNAs indeed expressed in the cells the authors are working with?

We used a list of never-essential genes that is being used in Cas9 dropout screens (Li et al., Genome Biology, 2015) but did not a priori check for actual expression these genes. The reviewer raises an important question, as expression of these controls is more important for CasRx screens than for Cas9 screens. To address the reviewer's concern we tested expression of the non-essential controls and found that only a small fraction is expressed, which indeed makes these controls inadequate for interrogating indiscriminate off-target cleavage. We are grateful for this critical comment, as it allowed us to now address this mistake.

First, we removed analyses interrogating the relationship between never-essentials and indiscriminate off-target effects from the manuscript. Instead, we use non-hit lncRNAs for this purpose: since a majority of lncRNAs targeted by the library are non-hit (of which a significant fraction is expressed), they can serve as internal controls for interrogating interference of possible indiscriminate off-target cleavage in conjunction with on-target cleavage. As shown in Fig. 4d and Supplementary Fig. 5e, gRNA LFC distribution curves targeting negative controls, non-expressed lncRNAs, and two groups of expressed lncRNAs [low-expressed ($0.01 < \text{TPM} < 1$) and high-expressed lncRNAs ($\text{TPM} > 1$)] precisely overlap. We conclude that targeting of expressed lncRNAs does not induce collateral RNA cleavage or adversely affect cellular fitness. The above is explained in the results section (page 6, lines 279-281). The data is shown in Figure 4d and in Supplementary Fig. 5e.

Second, to further prove this conclusion we performed fitness screens using a custom-library including 1634 non-targeting (NT) gRNA pairs as well as gRNAs targeting 300 always-essential (AE) genes and 300 never-essential (NE) genes (1200 gRNA pairs each). The NE genes that we have chosen now display robust expression but no fitness phenotypes in corresponding Cas9 or RNAi screens across CCLE cell lines (Tsherniak, et al., Cell, 2017) (Supplementary Fig. 1d,e). We compared the distribution of NT-gRNAs and gRNAs targeting NE genes in our screens across 3 cell lines. Since indiscriminate off-target cleavage is coupled to on-target cleavage, it would be expected in cells harboring NE-gRNAs, but not in cells carrying NT-gRNAs. Importantly, we did not observe differences in logarithmic fold-change (LFC) distributions between NE and NT gRNAs (Fig. 1j).

Third, to provide further verification of the absence of off-target cleavage following on-target cleavage in our cell lines, we performed knock-down experiments targeting GFP or an endogenous protein-coding gene in several cell lines, followed by RNA-seq. Indiscriminate off-target cleavage happens after on-target RNA degradation, during which Cas13 undergoes conformational changes exposing its catalytic domain, which then causes global RNA degradation. Our analyses detected only very few differentially expressed genes between the targeting and non-targeting conditions. These data provide further evidence for the lack of global gene dysregulation, which would be expected in scenarios of indiscriminate off-target cleavage (Fig. 1h,i).

The above is now explained in the results section (pages 3-4, lines 131-154). The data is shown in Figure 1h-j and Supplementary Fig. 1d,e.

B) What is the level of concordance between different gRNAs targeting the same lncRNA? The authors present the use of limited number of gRNAs per gene as an advantage because it allows for smaller

libraries, but there is also a shortcoming in that the confidence in individual hits (and in particular due to potential off-targets) is lower.

We agree on the importance of these questions. We performed analyses at three levels to show that our system does not suffer the shortcomings the reviewer is concerned about.

First, our design employs gRNA arrays (pairs), meaning that a lncRNA is targeted by two gRNAs in each cell. We use 2-7 arrays (4-14 gRNAs) to target each lncRNA family, depending on the lncRNA size/complexity. The rationale behind using arrays (rather than single gRNAs) was to target a broader range of isoforms and enhance the likelihood of each array functioning effectively. As suggested by the reviewer, we now show that the percentage of gRNA arrays contributing to the phenotype (fitness defect) of lncRNA hits and protein-coding gene hits is very high (between 87.7-96.5% in individual cell lines) (Supplementary Fig. 6c).

Second, we scaled the number of gRNA arrays based on the complexity (a parameter taking into consideration the length and the number of exons) of different lncRNA families. Only lncRNA families with extremely low complexity, characterized by an average of 1.4 exons and 573.3 bp length, are targeted by only 2 arrays. For these families the “gRNA sequence coverage” - representing the percentage of the lncRNA sequence targeted by its specific gRNAs - is up to 59.7%, with an average of 25.8% (Fig. 2e). Consequently, for such low-complexity lncRNAs, designing additional arrays encoding gRNAs that match our quality criteria (including predicted targeting efficacy and off-target filter) is largely unfeasible.

Third, To ensure that potential off-target lncRNAs are not enriched or retained among our hits due to their targeting by a limited number of arrays, we evaluated the proportion of lncRNA hits within each category (lncRNAs targeted by 2, 3, 4, 5, 6, or 7 arrays) after our off-target filtering. As expected, there is no evident distinction between lncRNA families targeted by 2 guides compared to those targeted by 3 and 4 guides. However, a moderate enrichment is noticeable for transcripts targeted by 5 guides, while a distinct enrichment is observed for those targeted by 6 and 7 guides (Reviewer fig. 1c). This observation is congruent with the notion that more complex lncRNAs are more likely to be functional and further support the notion that our hits are not enriched in off-targets.

The above is now explained in the results section (page 5, lines 200-202 and page 7, lines 309-314). The data is shown in Figure 2e and Supplementary Fig. 6c.

C) The current validation is based on few genes, mostly well-expressed lncRNAs with orthogonal support for their functions. A secondary library, containing other gRNAs targeting the dozens of other positive hits

in the screen using CasRx alongside additional shuffled controls can potentially further strengthen the results.

In response to the reviewer's suggestion, we extended our validation experiments in two ways:

First, we designed an additional library comprising new arrays targeting 93 of our previously identified hits, as suggested. We performed screens in three cell lines representing each entity (LN-18, A549, and MIA PaCa-2) and found highly significant correlations to the prior screens (Fig. 6a). Furthermore, we shuffled the controls as suggested: the new library has arrays targeting 230 entirely novel essential coding genes. Consistent with our prior screens, we observed a strong separation between the LFC distribution of the gRNAs targeting essential coding genes and the negative controls (Supplementary Fig. 1e).

Second, we generated new gRNAs against 16 lncRNA hits from our initial screen and performed FACS-based competition assays across three cell lines (LN-229, MIA PaCa-2, and NCI-H460). These experiments validated both common and specific lncRNA vulnerabilities across the cell lines (Fig. 6b).

The above is explained in the results section (page 8, lines 372-380), in more detail the methods section (pages 19-20, lines 942-966), and in figure legends (Fig. 6a,b). The data is shown in Figure 6a,b.

D) How many of the initial hits of the screen were removed by the “blacklisting” based on potential off-targets? If the authors use shorter kmers of 22 or 21 nts, are there additional suspected off-targets? Are these potential off-targets enriched with gRNAs where consistency between the different gRNAs targeting the same lncRNA is lower (i.e., is their correlation between confidence in no-off-targets and confidence derived from similarity between gRNAs)? These are key aspects to address the potential off-targets that are a major concern when, as the authors indicate, the rate of positive hits is lower than in traditional screens targeting protein coding genes, and in addition, many of the lncRNAs the authors are targeting are apparently expressed at extremely low levels in the cells they work with.

We should have been more precise and detailed on this in our initial manuscript, as the reviewer's comment is based on a misunderstanding.

To blacklist potential off-targets, we actually used 21 bp k-mers. More precisely, we used a 23 nt match allowing 2 mismatches at any position. This means by default that 21 bp perfect matches are included. By allowing 2 mismatches, we are more permissive in detecting off-targets as we allow for a larger potential “sequence pool” to align and be flanked as potential off-target (see example in Reviewer fig. 1d).

To validate this point, we conducted an in silico simulation experiment. We generated 100,000 random 23-mers, mapped them to the same reference used for off-target filtration, and allowed either 21 bp matches or 23 bp matches plus two mismatches. We subsequently counted the number of matches in both scenarios. As shown in Reviewer fig. 1e, when using 23 bp match allowing 2 mismatches we found 102 times more matches than using plain 21 bp. The threshold 21bp k-mer was mainly chosen for biological reasons: a minimum of 21 bp spacer length is essential for CasRx cleavage (Li et al, Nature methods 2021. Wessels et al, Nature biotechnology 2023). Using this approach, we blacklisted a total of 73 hits.

We clarify the above in the results section (pages 6-7, lines 289-290), in more detail the methods section (page 19, lines 899-9046).

E) In Figure 4a, the authors should show the distributions of the lncRNAs alongside the distributions of the positive and negative controls. Is there any difference between lncRNAs as a group and the negative controls?

We plotted these data (Fig. 4d, Supplementary Fig. 5e) as suggested. We answered the question in detail above (in response to Reviewer comment A) as follows:

“First, we removed analyses interrogating the relationship between never-essentials and indiscriminate off-target effects from the manuscript. Instead, we use non-hit lncRNAs for this purpose: since a majority of lncRNAs targeted by the library are non-hit (of which a significant fraction is expressed), they can serve as internal controls for interrogating interference of possible indiscriminate off-target cleavage in conjunction with on-target cleavage. As shown in Fig. 4d, Supplementary Fig. 5e, gRNA LFC distribution curves targeting negative controls, non-expressed lncRNAs, and two groups of expressed lncRNAs [low-expressed ($0.01 < \text{TPM} < 1$) and high-expressed lncRNAs ($\text{TPM} > 1$)] precisely overlap. We conclude that targeting of expressed lncRNAs does not induce collateral RNA cleavage or adversely affect cellular fitness

Minor comments:

A) Line 63: “because cis effects are splicing-independent” – it's not clear what is meant here, as lncRNA splicing was shown by several studies to be relevant to the activity of cis-acting lncRNAs.

We concur that in some cases, cis-acting lncRNAs may indeed rely on splicing. We have omitted this statement from the text.

B) Line 66: “lower numbers of true-positive dependencies than” – it is not clear enough what the authors mean by “dependencies” here. “Cellular phenotypes” would be better.

We changed the sentences as follows: “Finally, as discussed later in detail, double-strand break causation (DSBs) by Cas9 and related “toxicity” is a confounding factor that is particularly detrimental for the analysis of lncRNA screens, as there are fewer lncRNAs than protein-coding genes that cause cellular fitness phenotypes upon perturbation”. Please see introduction (page 2, lines 62-65).

C) Line 149: “genomic occupancy, sequence similarity, and transcript directionality” – this description is not clear, as the authors don’t really use sequence similarity, but overlap in genomic co-ordinates.

In the result section (page 4, line 170) we now clarify that for classifying isoforms into the same family, overlap of genomic occupancy is not sufficient. Isoforms should share at least one exon to be classified into the same family. We worked with genomic coordinates instead of comparing sequences only because the latter is the more complex computational approach to achieve similar results. As this results section is densely packed, limiting space for an exhaustive explanation of all technical details in the main text (Reviewer 3 in fact asked us to even shorten this part of the manuscript), we extensively address this point in the methods (page 15, lines 698-705) and in the Supplementary Fig. 2.

It is also unclear why they call these “lncRNA families” rather than “lncRNA genes”, as every family is essentially a gene with different transcript isoforms arising from it, even if the gene groups are not exactly the same as in GENCODE because of the slightly different grouping criteria.

We refer to “lncRNA families” because, as pointed out by the reviewer, our grouping criteria (which focus on defining potential functional units that need to be targeted individually) differ from those in GENCODE and other databases. Because of this, in some cases GENCODE genes can correspond to two or more lncRNA families or vice versa. To prevent confusion, we find it important to avoid the term “genes”.

D) Line 222: the authors should explain that the barcode is used for sample identification and multiplexing, its not clear from the text.

We included this information in the result section (page 6, line 246)

Reviewer Fig. 1: contains new information that has not been added to the manuscript. We provide these data to the reviewer because they support our line of reasoning to some of the specific questions. However, we did not add them to the manuscript as they are to some extent incremental to the new figures and partly also too granular.

Reviewer #2:

Remarks to the Author:

The manuscript by Montero et al. describes a high throughput strategy to leverage genome integrated CRISPR/CasRx for genome-wide depletion of IncRNAs in various cancers. They used transposons to insert multiple copies of CasRx into cancer genome with a gRNA library targeting 24,171 IncRNA families and

158,816 transcripts. Smaller scale of noncoding RNA screening has been demonstrated before and the main novelty of this work is the genome-wide scale of the screening. The study is overall interesting but a few concerns must be addressed

Major:

(1) Using transposons to deliver CasRx is highly efficient. However, the landing sites could be random which may lead to high variations in the screening. For example, the insertion could occur at different loci in different replicative experiments. Therefore, the screening results may not be reproducible due to synthetic impact by disrupted genes in adjacent to the insertions. This might be already reflected in their own analysis in Supplementary Figure 3b. i.e. Pearson correlation between replicative experiments is very low (equivalent to that between different cancer types, ~0.4-0.5). This type of variation is irrelevant to the expression level of CasRx gene but more likely by different genes affected by the insertion. The cells from replicative experiments can behave very differently even if the expression of CasRx is similar. The authors should carefully address the issue of variation and low reproducibility caused by random insertion. As a methodology paper, they should give a clear instruction on how to avoid such problem.

Clonal heterogeneity can in principle be a confounder in a CRISPR screen. We agree that this is an important consideration for any screen, as also discussed in detail in our recent review (Braun et al, Nature protocols 2022). Knowing this, we were extremely careful to avoid this problem. Below, we would like to show that equipping cell lines with genetic systems using transposons is not confounding the screening output.

First, our delivery approach relies on transient transposase expression, meaning that there is no ongoing transposon mobilization and mutagenesis in the clones used. In detail: We co-transfected the transposon vector carrying the CasRx expression cassette and the transposase vector using episomal vectors – enabling plasmid to genome mobilization of the CasRx cassette. After blasticidin selection, we expanded single clones to establish genetically homogenous cell lines with high CasRx activity. The episomal transposase vector gets eliminated during the process of antibiotic selection and clonal expansion as this plasmid is not auto-replicative. This setup prevents transposon mobility to different genomic locations, making it equivalent to cell lines established via lentivirus. To prove this point, we now performed PCR, showing that the HyPBBase transposase is “lost” in our clones (Supplementary Fig. 1b).

Second, transposon-aided genome-integration of CasRx does not significantly differ from lentiviral random genomic delivery, which is the more commonly used method for CRISPR screens. Although insertions of both approaches can in principle interfere with gene function, the likelihood is low, as exonic sequences

constitute only about 1% of the genome and – in addition - the likelihood of inserting in both alleles of a gene is essentially zero. To substantiate this point, we performed quantitative transposon insertion site sequencing (QiSeq), a method we developed earlier for highly-sensitive mapping of transposon insertions (Rad et al, Nature genetics 2015; Friedrich et al, Nature protocols 2017). The results show that none of the insertions in our clones is in lncRNA hits or essential coding genes (Supplementary Fig. 1c).

Third, knowing that cancer cell lines can be heterogenous, we use the exact same clone (derived from a single cell) for replicate screens. This ensures that we maintain genomic consistency to avoid a potential impact of heterogeneity on the screen. We now pinpoint this in the manuscript, giving a clear instruction to readers, as suggested by the reviewer.

In summary, transposon-based CasRx delivery is not a source of replicate differences. The Pearson correlations between replicates are within an expected range, which is related to the intrinsic nature of any screen with low numbers of hits (which is a major difference to genome-scale perturbation of the protein-coding genome). We will comment on this aspect in detail below (major point 3).

The above is now explained in the results section (page 3, lines 110-113 and lines 118-124), in more detail the methods section (page 15, lines 712-739). The data is shown in Supplementary Fig. 1b,c.

(2) The design of gRNA is not clear. For example, (a) how the pair of gRNAs targeting the same transcript is chosen?

As suggested, we added more details to the description of how the gRNA arrays were designed.

In the results section, the description is as follows (page 5, lines 196-205):

“First, we exploited CasRx’s capability to mature its own gRNA array³⁴ and designed each vector to encode a pair of non-redundant gRNAs targeting the same lncRNA family. Second, we scaled the number of gRNAs according to lncRNA complexity (as defined by lncRNA lengths multiplied by the number of exons) by selecting 2-7 gRNA arrays (4-14 gRNAs) for each family, depending on the corresponding lncRNA length and the number of exons (Fig. 2d). Third, we developed an iterative gRNA design process, with each cycle having less strict requirements to (i) gRNA quality (as a predictor of cleavage efficiency⁴⁷) and (ii) gRNA distribution along each lncRNA family (as a proxy to capture isoform complexity). Iterations were repeated until design criteria were fulfilled. lncRNA families for which minimal criteria could not be met were removed from the list (Fig. 2a,c) (see methods).”

In the methods section the revised description is as follows (page 17, lines 800-816):

“We designed the arrays using four subsequent rounds of decreasing stringency (lncRNA families whose design was not accomplished in the previous round were processed in the following round) according to three parameters: gRNA quality (Standardized Guide Score SGS⁴⁷); a minimum distance between gRNAs in the same array (defined as Minimum Local Distance MLD); a minimum distance between any gRNA targeting the same family (defined as Minimum Global Distance MGD). First, round, $SGS \geq 0.75$; $MLD \geq 30\%$ of the lncRNA family length; $MGD \geq 10\%$ of the lncRNA family length. Second round, SGS in the best quartile; $MLD \geq 25\%$ of the lncRNA family length; $MGD \geq 5\%$ of the lncRNA family length. Third round, SGS in the best quartile; $MLD \geq 10\%$ of the lncRNA family length; $MGD \geq 1\%$ of the lncRNA family length. Fourth round, SGS any value; $MLD \geq 10\%$ of the lncRNA family length; $MGD \geq$ no overlap.

In each round, we selected the gRNA in position 1 for each array as follows: First, the gRNA with the best SGS was assigned to position 1 of the first array. For position 1 of the second array, we selected (of the remaining designed gRNAs) the one with the best SGS that fulfilled the MGD criteria. The same process was repeated for all the remaining arrays. We choose the gRNA in position 2 for each array as follows: For the first array, we picked (of the remaining gRNAs) the one with the best SGS that satisfied both the MGD and MLD criteria. This process was iterated for all arrays. If, at any point during this process, no gRNA meeting the SGS, MGD or MLD criteria was found, the lncRNA passed to the next less stringent design round.”

Reviewer comment: (b) The number of gRNA pairs only depends on the number of exons, instead of the length and secondary structure of the transcript, which I feel are more important features to be considered.

This is a misunderstanding. We agree with the reviewer that the lncRNA length is an important feature, which was explicitly considered during our design. In detail: We used both parameters (length and number of exons) to define the “complexity” value, which was used to scale the number of gRNA arrays per lncRNA family. The number of exons was considered in order to better capture (and target) isoform diversity (lncRNAs with higher numbers of exons tend to have more isoforms).

These details are now more strongly emphasized throughout the manuscript, including the results section (page 5, line 201; page 5, lines 226-227.), the methods section (page 16, line 790), and in Fig. 3d.

Reviewer comment: They should perform the screening in the same cells with an independent set of gRNAs and compare the results. Further analysis of the difference between the two independent screenings may provide valuable information on the impact of gRNA coverage/design.

As suggested, we designed an additional library comprising new arrays targeting 93 of our previously identified hits. We performed screens in three cell lines representing each entity (LN-18, A549, and MIA PaCa-2) and found highly significant correlations to the prior screens (Fig. 6a). Furthermore, we shuffled the controls as suggested: the new library has arrays targeting 230 entirely novel essential coding genes. Consistent with our prior screens, we observed a strong separation between the LFC distribution of the gRNAs targeting essential coding genes and the negative controls (Supplementary Fig. 6e).

Furthermore, we generated new gRNAs against 16 lncRNA hits from our initial screen and performed deep FACS-based competition assays across three cell lines (LN-229, MIA PaCa-2, and NCI-H460). These experiments validated both common and specific lncRNA vulnerabilities across the cell lines (Fig. 6b).

These results further validate at different levels the robustness of our screening design.

The above is now explained in the results section (page 8, lines 372-380), in more detail the methods section (pages 19-20, lines 942-966), and in figure legends (Fig. 6a,b). The data is shown in Figure 6a,b and Supplementary Fig. 6e.

(3) Delivery of gRNA to cell lines is always a critical parameter in screening work. The authors failed to show the coverage of gRNAs in each screen experiment as a very essential quality control. Not every lncRNA could be screened if the coverage in some cell line is low.

We agree that the even representation of the gRNA library is a critical parameter for conducting a screen. Its key determinants are (a) the quality of the library (distribution of gRNAs) and (b) the delivery and coverage (number of cells per gRNA). We explain in detail below that we applied the highest design and experimental standards to ensure that both aspects are fulfilled:

(a) Quality of the library. Fig. 3a shows Lorenz curves of gRNA representation of our Albarossa library and for other widely used CRISPR libraries. With an area under the curve of 0.59 the distribution of our library is superior to the other libraries. Moreover, to mitigate the possibility of false positive drop-outs arising from gRNAs with low coverage, we excluded the gRNAs with low representation in the original library during the analysis process. This step, which is a common procedure for CRISPR screens, is described in the method section (page 18, line 894).

(b) gRNA delivery and coverage. Our gRNA transduction protocol was explicitly designed to guarantee adequate experimental coverage. For each cell line, we first performed individual titration experiments to calculate the viral titer needed to achieve a survival rate of 25% after antibiotic selection. When conducting the screens, we used the relevant viral titers for library transduction and confirmed survival rates of 25%

survival in the screening setting. For each screen, we transduced 94 Mio cells, which - at the MOI of 0.25 – results in a coverage of 300 cells per gRNA array, which is in the range of what is commonly used for CRISPR screens (e.g. DepMap screens deploy a lower coverage of 100).

This coverage is adequate and supports high-quality screening, as shown by the gRNA distribution curves of the screens, which display homogenous normal distribution (Supplementary Fig. 4b). Only few gRNA arrays display low sequencing read counts, and – as expected – gRNAs targeting always-essential control genes constitute 50% of this group, whereas they only represent 1% of the library. In contrast, none of the negative control gRNAs fell into this category.

The above is now explained in the results section (page 6, lines 248-252; page 6, lines 259-264), in the methods section (pages 18, lines 864-868). The data is shown in Figure 3a and in Supplementary Fig. 4b.

Reviewer comment: This may also explain the low Pearson correlation in Supplementary Fig. 3b. Furthermore, transduced cells before antibiotic selection were collected as control samples (e.g. Liu, Horlbeck et al. 2017, Liu, Cao et al. 2018). It is rational to involve such control to calculate the LFC.

We would like to answer this question by commenting on two aspects separately:

1. LFC normalization approach. LFC normalization using transduced cells before antibiotic selection is uncommon as it substantially increases costs and efforts. Commonly used as a baseline are either transduced cells after selection (Zhu et al, Nature biotechnology 2016. Liu et al, Science 2017. Liu et al, Nature biotechnology 2018) or the library itself (Shalem et al, Science 2013. Tsherniak et al, Cell 2018). Both approaches have been shown to be equivalent (Shalem et al, Science 2013). We chose the “library-approach”, which eliminates the necessity to generate and sequence a separate NGS library for each screen - and hence supports our aim to describe a scalable and cost-effective method for pan-cancer applications.

2. Pearson correlations. The Pearson correlations between replicates in our lncRNA screens are within an expected range. Replicate correlations in lncRNA screens are lower than for genome-wide perturbations of the protein-coding genome. This is due to the substantially smaller number of hits (gRNAs dropping out) in lncRNA screens. While perturbations with fitness effects are subjected to selective pressure (resulting in a greater correlation between replicates), gRNAs without biological effects have more random distribution (caused by experimental noise).

To demonstrate this notion, we calculated Pearson correlations between screening replicates using LFC values of essential coding genes that are known to have biological effects. As shown in (Supplementary

Fig. 4e), the Pearson correlations between replicates are very high for these essential coding genes (0.89-0.93). Similarly, lncRNA hit replicates are highly correlated. In contrast, negative controls (no biological relevance) display much lower correlation, as expected (Supplementary Fig. 4e).

As a second line of evidence, we reanalyzed previous lncRNA screens (Zhu et al, Nature biotechnology 2016. Liu et al, Science 2017. Liu et al, Nature biotechnology 2018). As expected, our experiments exhibit comparable replicate correlations to those screens (Supplementary Fig. 4d). The data thus further argue against a confounding role of our LFC normalization approach (not least because those studies employed transduced cells after selection for baseline calculation).

The above is now explained in the results section (page 6, line 265-266), in more detail in the supplementary figure legends (Supplementary Fig. 4e). The data is shown in Supplementary Fig. 4d-e.

(4) The inclusion of lncRNAs in the screening mostly depends on their expression level in various cancer cells. Even though they show in figure 3 the enrichment of functional traits in the targeted lncRNAs, it remains an issue that some modestly expressed essential lncRNAs are missing from the screening. What would be the limiting factor for the capacity of the screening? Why not include all? They should at least do saturation screening in one cancer cell type and compare the results with the Albarossa library to show if any essential one is missing.

Our screening approach indeed supports saturation screening. To prove this, we designed and conducted a saturation screen (see below). However, the main aim of our study was to create a lncRNA screening platform that can be used for large-scale pan-cancer studies. Such large-scale cross-entity mapping of lncRNA vulnerabilities is highly relevant because of a series of reasons (discussed in more detail in the discussion section (pages 9-10, lines 436-462)). For example, pan-cancer mapping of lncRNA dependencies will be a decisive step towards systematic context-based inference of lncRNA function.

Why can the pan-cancer approach not be easily achieved by saturation screening? The key issue with saturation screens is the lacking scalability – due to the extensive size of the lncRNA transcriptome. The number of lncRNA genes vastly exceeds the number of protein-coding genes. Targeting all 97,817 lncRNA families requires a library >4 times larger than Albarossa. Not only the experimental efforts (starting with transduction of 375 Mio cells), but also the technical challenges for conduction of such screens are enormous. For most laboratories, even single screens using such a huge library are not feasible, let alone comparative screens across cell lines.

One possibility to address this problem would be to create cell-line specific smaller libraries targeting only lncRNAs expressed in individual cell lines. This would mean however, that libraries would need to be individually designed, purchased (each ~10,000 USD), cloned and tested for each cell line. Furthermore, comparative analyses across screens would be very difficult because using different libraries is the main source of batch effects (Dempster et al, Nature communication 2019).

Our solution to these critical bottlenecks was to develop a genome-scale library that is experimentally manageable and at the same time strongly enriched with functional lncRNA targets. Its rational design incorporates target prioritization based on expression, evolutionary conservation, and tissue-specificity, thereby reconciling high discovery-power and pan-cancer representation with scalable experimental throughput. We show that the screening platform identified numerous context-specific and common-essential lncRNAs – enabling for the first time systematic pan-cancer exploration of lncRNA biology in health and disease.

To confirm that Albarossa library is enriched in functional lncRNAs, we followed the reviewer's suggestion and designed a library targeting expressed lncRNAs in the LN-18 cell line (TPM > 0.01) that were not included in the Albarossa library (to achieve saturation mutagenesis). We conducted a screen on the LN-18 cell line using this library (Supplementary Fig. 6d,e) and we identified 26 lncRNA hits (Figure 4g). In comparison, the Albarossa screen identified 2.5 times more hits (Figure 4f). Subsequently, we calculated the discovery rate for both libraries (percent of hits among expressed lncRNA families targeted). This discovery rate of the Albarossa library was 2 or 3 times higher for all hits or high-confidence hits, respectively (Figure 4h). Thus, by targeting only 1/4 of the all lncRNAs with our Albarossa library, we detected 2/3 of all hits or 3/4 of all high-confidence lncRNA vulnerabilities.

In summary, whilst we show that our screening approach supports saturation screening, pan-cancer screening requires a different methodology – and we demonstrate that our approach reconciles high discovery-power and pan-cancer representation with scalable experimental throughput.

The above is now explained in the results section (page 7, lines 298-307). The data is shown in Fig. 4g,h and Supplementary Fig. 6d,e.

(5) CasRx is a class 2 type VI CRISPR-Cas RNA endonuclease. That means the targeted RNA is not necessarily degraded by the enzyme lacking exonuclease activity. The RNA may be simply cut into two fragments. This is not a problem when screening coding gene as cutting off the start codon is adequate to suppress the mRNA translation. However, the targeted noncoding RNA may still be partially or fully

functional, causing high ratio of false negative. Comparison of the screening result with the ASO-based screening will give us clear clue about this concern.

ASO-based screening has two main limitations. ASOs cannot be encoded genetically as they are DNA-based. Their incorporation into pooled screening systems relying on genome-integrated permanent barcoding and long-term fitness readouts is therefore impossible. Moreover, the outcome of ASO-based targeting is similar to the one of CasRx: cutting into two pieces. In brief, ASOs function via RNase H, a non-sequence-specific endonuclease enzyme that degrades RNA/DNA hybrids. Like CasRx, RNase H lacks exonuclease activity. While this enzyme naturally has the potential to degrade RNA hybridized with DNA, the ASO-design confines hybridization (and degradation) to few bases. Thus, ASOs cleave the targeted RNA into two fragments, like CasRx.

We do however agree with the reviewer's point that cleaved RNAs can still function. We should have pointed out more clearly that in fact, we explicitly designed our perturbation system to minimize this problem. We utilize gRNA arrays targeting two distinct regions of the lncRNA in each cell, which enhances the ability to completely inactivate the specific lncRNA transcript. Moreover, each resulting fragment lacks either the 5' RNA cap or the poly-A tail. These two structures are crucial for maintaining RNA stability. Without them, the remaining fragments are highly susceptible to degradation by endogenous RNA exonucleases. We now explain these design considerations.

(6) Due to the inherent problem of CasRx discussed above, using the same enzyme (CasRx) for validation may not be appropriate (Fig. 5). The authors may want to use orthogonal methods for validation of their findings. For example, ASO-based, RNAi-based and CRISPRi-based knockdown could be considered. In addition, rescue experiment can be performed to confirm that the phenotype was indeed caused by lncRNA deficiency.

We now point out more clearly that 24.4% of our hits (50 lncRNAs) have already been studied and validated using the orthogonal methods referred to by the reviewer (ASOs, RNAi or CRISPRi) (results section (page 350-351)). These data – together with our extensive CasRx-based validation efforts (see also new data above, new Figure 6a-b) - serve as robust evidence for the validity of our screens.

(7) Proliferation is only one characteristics of cancer. The lncRNAs screened by the current design may not be cancer specific. Authors should consider using some normal cells as background controls, and they may identify some essential genes for general cell proliferation (which is less interesting to this study purpose). Furthermore, the authors could consider screening metastasis, or other cancer hallmark feature related lncRNAs.

We feel that these questions are far beyond the scope of the manuscript, which is to provide proof-of-concept CasRx lncRNA fitness screening in and across cancer cell lines. It took us many years to achieve this aim and develop the methods/protocol for cancer cell lines. Developing a CasRx-based screening system in primary healthy cells is an enormous task that comes with numerous new experimental obstacles. For example, achieving high-level CasRx expression relies on transfection and transposon-based delivery of large constructs. Primary healthy cells are difficult to transfect, and establishing single cell clones is yet another challenge. These and many other technical hurdles make it impossible to address this suggestion within reasonable time frames (even when using the streamlined protocol for cancer cell lines, equipping cells with CasRx and conducting the screens takes a minimum of 5 months). The same is true for screens related to other cancer hallmarks, such as metastasis, which is an organismal phenotype. Metastasis screens would require the development of complex in vivo assays. The animal license applications alone would take us more than a year to obtain.

(8) The potential impact of the genome-integrated CasRx protein in human cancer cell lines. However, it has been reported that CasRx protein has collateral activity, which is detrimental to cell proliferation (Shi, Murphy et al. 2023). It seems rational to apply an inducible system to control the CasRx protein expression to prevent the side effects from the constitutive expression of CasRx proteins. In addition, the authors should carry out RNA-seq assay in those transgenic and parental cell lines with or without target gRNA transduction to figure out differential genes may be involved in function lncRNA screening.

Collateral perturbation can be a confounder of a screen, and we agree that this possibility has to be excluded, which we did at multiple levels. The related experiments are shown and discussed below (point c). Before, we would like to briefly comment (and present new data) on two related aspects.

(a) CasRx collateral cleavage can only happen in conjunction with on-target RNA degradation, during which Cas13 undergoes conformational changes exposing its catalytic domain - which then causes global RNA degradation (Zhang et al, Cell 2018). Because of this, without the presence of on-target gRNA the CasRx protein cannot show such unwanted effects. Thus, the use of an inducible system is not suitable to address potential CasRx collateral perturbation. Nevertheless, we performed RNA-Seq in the parental cell culture and related CasRx transgenic cell clones, as suggested by the reviewer. As expected, PCA analyses show that CasRx clones and their parental lines cluster together (Fig. 1c). Furthermore, their expression correlations are extremely high (>0.981), with only minimal differences between parental cultures and their descendant clones (some differences are expected for clonally expanded single cells) (Supplementary Fig.

1a). If anything, there is a slight trend towards higher expression of some genes in CasRx-transgenic lines, which is the opposite of what would be expected in case of global indiscriminate CasRx activity.

(b) When first described in bacteria, Cas13 systems were reported to display off-target effects due to indiscriminate RNA degradation (Abudayyeh et al, Science 2016. East-Seletsky et al, Nature 2016). Such effects had not been observed in initial reports using CasRx in mammalian cells (Konermann et al, Cell 2018. Wessels et al, Nature biotechnology 2020), but have been recently described in connection with very high episomal CasRx expression (Ai et al, Nucleic Acid Res. 2022. Shi et al, Commun. Biol. 2023). In our system CasRx is expressed from genome-integrated positions. We do not face the problem of having too much CasRx expressed. The challenge for our approach was rather to achieve high-enough CasRx activity.

(c) We ruled out the possibility of collateral CasRx activity in our system using three different approaches:

First, since a majority of lncRNAs targeted by the library are “non-hit” (of which a significant fraction is expressed), they can serve as internal controls for interrogating interference of possible indiscriminate off-target cleavage in conjunction with on-target cleavage. As shown in Fig. 4d and Supplementary Fig. 5e, gRNA LFC distribution curves targeting negative controls, non-expressed lncRNAs, and two groups of expressed lncRNAs [low-expressed ($0.01 < \text{TPM} < 1$) and high-expressed lncRNAs ($\text{TPM} > 1$)] precisely overlap (Fig. 4d, Supplementary Fig. 5e). We conclude that targeting of expressed lncRNAs does not induce collateral RNA cleavage or adversely affect cellular fitness in any of the cell lines.

Second, to further prove this conclusion we performed fitness screens using a custom-library including 1634 non-targeting (NT) gRNA pairs as well as gRNAs targeting 300 always-essential (AE) genes and 300 never-essential (NE) genes (1200 gRNA pairs each). The NE genes chosen display robust expression but no fitness phenotypes in corresponding Cas9 CRISPR screens across CCLE cell lines (Tsherniak, et al., Cell, 2017) (Supplementary Fig. 1d,e). We compared the distribution of NT-gRNAs and gRNAs targeting NE genes in our screens across 3 cell lines. Since indiscriminate off-target cleavage is coupled to on-target cleavage, it would be expected in cells harboring NE-gRNAs, but not in cells carrying NT-gRNAs. Importantly, we did not observe differences in logarithmic fold-change (LFC) distributions between NE and NT gRNAs (Fig. 1j).

Third, as suggested by the reviewer, we performed knock-down experiments targeting GFP or an endogenous protein-coding gene in several cell lines, followed by RNA-seq. Differential expression analysis between the targeting and the non-targeting conditions showed a lack of global gene dysregulation, which would be expected in scenarios of indiscriminate off-target cleavage (Fig. 1h,i).

Altogether, these data provide overwhelming evidence for a lack of indiscriminate collateral cleavage in our systems. Our screening setting thus achieves an optimal window of functionality - characterized by sufficiently high CasRx on-target activity but no indiscriminate off-target RNA degradation.

The above is now explained in the results section (pages 3-4, lines 131-154; page 6, lines 279-281). The data is shown in Fig. 1c,g-j, Fig. 4d, Supplementary Fig. 1a,d,e and Supplementary Fig. 5e.

Minor:

(9) Figure 5g, is the statistics (mean, SD and P values) from repeated quantification or independent experiments?

This information is provided in the figure legends (Fig. 6). The statistics are derived from independent experiments.

(10) In Data Availability, hyperlink to the data reservoir should be provided.

The data have been uploaded to ENA and access codes have been generated. The hyperlink is now available in the data availability section (page 10, line 476). However, please note that the link will only become functional after the manuscript is accepted, at which point the data will be made public.

(11) The numeric value of y-axis in Supplementary Figure 4f should be 150, not "1.50".

We have changed the numeric value. This panel can be found as Figure 1g of the revised manuscript.

(12) Line 545, should be HEK-293T, not "HEK-2937T".

We have corrected this misspelling. Methods section (page 13, line 626).

Reviewer #3:

Remarks to the Author:

The manuscript by Montero et al. reports a novel Cas13/CasRx-based screen to systematically and unbiasedly identify lncRNAs with potentially important functions in multiple cancer cell lines. The authors developed a genome-integrated CasRx system using different cancer lines to achieve genome-scale knock-down of lncRNAs. As depletion of lncRNA transcripts is a challenging task, the study presents an elegant system to screen for lncRNA loss of functions, overcoming limitations of several other studies. I believe it is an important and well-designed study; the results of the screen will be of the interest for the lncRNA community for further follow-up analyses. The manuscript is very well-written. My specific

comments to clarify some aspects of the study and improve its readability are summarized below.

Major comments on data presentation:

1. I found that there are too many technical details in the results describing library design, generation, sequencing and mapping lncRNA dependencies. These sections can be shortened and repetitive technical details can be moved to the material and methods. Shortening these parts would improve the readability and accessibility of the manuscript.

As suggested, we have tried to condense the relevant paragraphs in the results section and moved some details to the method section. We were however somewhat constrained as at the same time reviewers 1 and 2 asked for some more details that we had to incorporate. Changes in the text can be found in the results section (page 3, line 110; page 5, line 202; page 6, line 248).

2. The authors should elaborate on the fitness screening and give details of their assays and readouts. Currently, the authors only cite publications that used the same or similar approaches.

To address the reviewer's suggestion, we now explain in more detail the principles of the fitness screening procedures and analyses in the results section (page 6, lines 254-256), in the methods section (page 19, lines 906-910) and incorporated a scheme of the experimental workflow into Fig. 4a.

3. The functional data presented in Figure 5 should be expanded and explained in the results section. Description of cellular assays used for functional assessment and validation of screen hits is missing. They should be mentioned in results and included in the method section.

We now describe in more detail the proliferation assays, colony formation assays, as well as new competitions assays performed for the revision of the manuscript in the results section (page 8, lines 372-382) and in the figure legends of Fig. 6a,b. Moreover, we added relevant technical details to the methods section (pages 19-20, lines 942-966).

4. In my opinion, what is missing is a figure illustrating the overview of the screen. The experimental design of the screen gets lost in technical details of individual steps. I would recommend to introduce a nice scheme including a summary of cell lines which were used, the constructs integrated and/or transfected to perform the actual screen, what read-outs were used etc.

We agree with the reviewer's view that a scheme would improve the understanding of our screening method. As suggested, we have included a figure providing an overview of the experimental workflow and

components (including cell lines, vectors) as well as key technical details (transduction MOI, screening coverage, and read-outs). (Fig. 4a)

Further specific comments:

5. Optimization of a genome-integrated CasRx system

5.1 The authors should list the CasRx sequence in supplemental files and also clarify if the NLS was added to the sequence.

We are grateful to the reviewer for pointing out that the scheme in figure 1a is misleading. The CasRx protein used in our study is indeed fused with two NLS sequences, positioned at the C-terminus and N-terminus. We adapted the scheme accordingly and now also explicitly mention the importance of the NLS on of the results section (page 3, line 111; page 7, line 309), on the discussion section (page 9, lines 409-413) and on Fig. 1a. Plasmid sequences are available in Supplementary Table 10.

5.2 From the description of the genome-integrated CasRx system, it was not clear if the system would work for both, nuclear and cytoplasmic transcripts.

We clarify that the CasRx system has an NLS, supporting nuclear RNA cleavage. Because any RNA is produced in the nucleus and only few have pure cytoplasmic localization, the nucleus is the most critical compartment. The ability of CasRx to translocate and cleave RNA in the nucleus is a main advantage in comparison to RNAi based approaches.

To further substantiate this point, we also performed nuclear/cytoplasmic fractionation followed by RNA-Seq. The results will be explained below in response to comments 5.3 and 6.1.

5.3 The authors should add small-scale control experiments showing that their CasRx system is efficient for depletion of nuclear (ie. Malat1) and cytoplasmic lncRNAs. A few abundant transcripts would be sufficient.

To quantify nuclear and cytoplasmic lncRNA, we performed nuclear/cytoplasmic fractionation followed by deep full-length RNA-Seq for all CasRx clones. As expected, we found enrichment of numerous lncRNAs in the nuclear fraction. Examples are well-known nuclear lncRNAs, such as MALAT1 and NEAT1. In contrast, enrichment of transcripts in the cytoplasmic fraction is more rare, consistent with previous reports (Zaghlool et al, Scientific reports, 2021) (Fig. 5a).

To demonstrate that CasRx is targeting nuclear lncRNAs, we individually validated eight lncRNA screening hits that are enriched in the nuclear fraction. This selection includes very abundantly expressed candidates, such as MALAT1 and NEAT1. We already validated the effects of NEAT1, MIR222HG, Lnc-AKAP12-1, and FENDRR in the initial version of the manuscript by quantifying proliferation and colony formation capability of cells. We now expanded such functional studies to the entire set of selected nuclear lncRNAs (MALAT1, LINC-PINT, LINC00824, and Lnc-UTRN-3) by performing FACS-based competition assays. The results confirm that CasRx supports targeting of nuclear lncRNAs (Fig. 6b). In addition, we also validated two lncRNAs that show cytoplasmic enrichment (CYTOR and SNHG16) (Fig. 6b).

The above is now explained in the results section (page 8, lines 372-380), in more detail the methods section (pages 19-20, lines 942-966), and in figure legends (Fig. 6a,b). The data is shown in Figure 6a,b.

6. Validation of lncRNA dependencies

6.1 The authors should mention what is the proportion of nuclear vs. cytoplasmic lncRNA transcripts which were identified in the screen (Figure 5d). Is there a bias of the screen results towards nuclear enriched transcripts?

To address this question, we intersected our screening data with RNA-Seq results of the fractionation experiments described above. We found that 7.4% of lncRNAs targeted by our library display nuclear enrichment, whilst 18% of screening hits are lncRNAs with enrichment in the nuclear fraction. In addition, 0.6% of lncRNAs targeted in Albarossa shows enrichment in the cytoplasmic fractions, whereas 3.4% of our hits display cytoplasmic enrichment. These data indicate that (i) compartmentalization of lncRNAs indicates functionality and that (ii) our screening approach enables detection of both nuclear and cytoplasmic lncRNA essentialities (Fig. 5a).

The above is now explained in the results section (page 7, lines 309-314). The data is shown in Fig. 5a.

6.2 For the selected six lncRNAs, the authors should demonstrate changes in their expression levels upon CasRx-depletion by a simple qRT-PCR or similar methods.

As suggested, we conducted qRT-PCR assays confirming CasRx-induced lncRNA expression changes. These results are shown in (Fig. 6e).

Decision Letter, first revision:

Dear Roland,

Thank you for submitting your revised manuscript "Genome-scale pan-cancer interrogation of lncRNA dependencies using CasRx" (NMEMH-A51928A). It has now been seen by the original referees and their comments are below. The reviewers find that the paper has improved in revision, and therefore we'll be happy in principle to publish it in Nature Methods, pending minor revisions to satisfy the referees' final requests and to comply with our editorial and formatting guidelines.

TRANSPARENT PEER REVIEW

Nature Methods offers a transparent peer review option for new original research manuscripts submitted from 17th February 2021. We encourage increased transparency in peer review by publishing the reviewer comments, author rebuttal letters and editorial decision letters if the authors agree. Such peer review material is made available as a supplementary peer review file. Please state in the cover letter 'I wish to participate in transparent peer review' if you want to opt in, or 'I do not wish to participate in transparent peer review' if you don't. Failure to state your preference will result in delays in accepting your manuscript for publication.

ORCID

Sincerely,
Madhura

Madhura Mukhopadhyay, PhD
Senior Editor
Nature Methods

Reviewer #1 (Remarks to the Author):

The authors have extensively changed the text and provide both new data and new analysis, that makes their case that the CasRx-based library screening is indeed a powerful and accurate method for identifying functionally relevant lncRNAs in cancer cells.

Since the text and figures have changed quite substantially, I have some additional changes that I would like the authors to make, in particular since the paper is likely to set a standard for future papers in the field:

1. I still really dislike the use of the term “lncRNA families” for simply overlapping lncRNA transcripts, which in most cases correspond to a single GENCODE gene, and sometimes correspond to two overlapping GENCODE genes. “Families”, in the context of both proteins and miRNA, correspond to distinct loci, typically far apart and on different chromosomes, that encode transcripts with similar sequences/functions. This is not the case here, therefore I again ask the authors use the term “genes”, which is commonly used in the context of genome annotation to refer to a collection of overlapping transcripts (which often have different promoters/introns etc.), or to use some term like “mega-genes” or “gene-set”, and not the confusing “family” terminology. If they want to distinguish their “genes” from GENCODE genes, they can use the term ENSGs to refer to GENCODE genes.
2. Line 198 “complete inactivation” sounds deterministic “effective inactivation” is better.
3. The threshold of >0.01 TPM needs to be explained better. It seems that in the universal analysis (Supp. Fig 3b), the authors are using the median expression across the CCLE when 0.01 is used as a threshold, but this is not explained in the text that it is the `_median_` across the cell lines (line 213). Since lncRNAs are typically quite tissue specific, it doesn’t make any sense to look at this median, since expression is close to 0 in most cell lines. I understand that the library is not going to be re-designed, and the authors need to explain their choices retrospectively, but it should be clearly explained in the text (around line 213) that this threshold is the median across all the cell lines, and it should be explained in

the Discussion that future libraries may want to use other criteria. The authors then still use this insensible 0.01 TPM threshold in Figure 3a, which is not doing a universal analysis but rather looks per-cell-line. In the rebuttal, they seem to agree that TPM of 0.1 is a much more reasonable threshold; I don't understand why they still use 0.01 per cell line in Fig. 3a. The justification seems to again come from the analysis that looks at the `_median_` across all the CCLE, which doesn't make sense. Fig. 3a should show TPM>0.1 or TPM>1 results, not TPM>0.01, which doesn't make any sense in a single cell line.

4. Lines 226-227, I don't understand what lncRNA complexity has to do with lncRNA secondary structure. One refers to the complexity of the splicing, and another to the folding of the RNA. Longer RNAs have more potential to have some structured regions, but the sentence in current form is confusing.

5. For the consistency between the effects of different gRNA arrays targeting the same lncRNA, the authors currently show in Fig. 6c the % of gRNAs that contribute to the phenotype, which is a confusing number (I don't quite understand what this means). A more intuitive metric would be to compute per-array log₂fc and then the authors can compare these between gRNA arrays targeting the same gene vs. those targeting different genes.

6. Line 300: "saturation mutagenesis" is a term used to refer to mutations of all possible bases in a sequence. Here, this is not what the authors are doing (in fact, they are not doing any mutagenesis of all), therefore, "more exhaustive/comprehensive targeting" would be more appropriate here.

7. Figure 5c – the terminology of "off-targets" is confusing, as the authors don't know that these are off-targets, these can be merely suspected to be off-targets because they are close to other genes. "% close to other genes" is more accurate.

8. Fig. 5f-g – statistical significance should be reported.

Reviewer #2 (Remarks to the Author):

The authors have adequately addressed most of my previous concerns and the work looks much more consolidated than the previous submission. This study provides a very useful platform/resource to screen functional lncRNAs in various characteristics of cancer. The following comments may help to further clarify the manuscript.

1. For all supplementary tables, legends should be provided to clarify the contents; e.g. in Supplementart Table 4, I'm not sure if everyone understands what these transcript IDs correspond to?

Are these ID from ENSEMBL or what? in Supplementary Table 1, these contents make no sense without proper format and legends.

2. The authors claimed in Line 199 that "the 5' RNA cap or the poly-A tail - structures that are essential for maintaining RNA stability". There's no evidence to support this. Can RNA-seq genome browser track for some examples be shown to reveal the degradation of some targeted lncRNAs?
3. To respond to my previous point 6, the authors claimed that 24.4% (50 lncRNAs) were validated by orthogonal methods. Apparently, they meant by other studies. However, there's no citation or list of these 50 lncRNAs. Where is this number from?
4. Will the authors consider sharing these resources in public reservoir (e.g. Addgene) to benefit more users?

Reviewer #3 (Remarks to the Author):

In the revised version of the manuscript, the authors substantially improved manuscript readability and sufficiently addressed my concerns and comments. I believe the manuscript is now ready to be published.

Author Rebuttal, first revision:

Reviewer #1 (Remarks to the Author):

The authors have extensively changed the text and provide both new data and new analysis, that makes their case that the CasRx-based library screening is indeed a powerful and accurate method for identifying functionally relevant lncRNAs in cancer cells.

Since the text and figures have changed quite substantially, I have some additional changes that I would like the authors to make, in particular since the paper is likely to set a standard for future papers in the field:

1. I still really dislike the use of the term "lncRNA families" for simply overlapping lncRNA transcripts, which in most cases correspond to a single GENCODE gene, and sometimes correspond to two overlapping GENCODE genes. "Families", in the context of both proteins and miRNA, correspond to distinct loci, typically far apart and on different chromosomes, that encode transcripts with similar sequences/functions. This is not the case here, therefore I again ask the authors use the term "genes", which is commonly used in the context of genome annotation to refer to a collection of overlapping transcripts (which often have different promoters/introns etc.), or to use some term like "mega-genes" or "gene-set", and not the confusing "family" terminology. If they want to distinguish their "genes" from GENCODE genes, they can use the term ENSGs to refer to GENCODE genes.

As suggested by the reviewer, we changed the term "lncRNA family" and use "lncRNA gene".

2. Line 198 "complete inactivation" sounds deterministic "effective inactivation" is better.

We changed the term as suggested.

3. The threshold of >0.01 TPM needs to be explained better. It seems that in the universal analysis (Supp. Fig 3b), the authors are using the median expression across the CCLE when 0.01 is used as a threshold, but this is not explained in the text that it is the `_median_` across the cell lines (line 213). Since lncRNAs are typically quite tissue specific, it doesn't make any sense to look at this median, since expression is close to 0 in most cell lines. I understand that the library is not going to be re-designed, and the authors need to explain their choices retrospectively, but it should be clearly explained in the text (around line 213) that this threshold is the median across all the cell lines, and it should be explained in the Discussion that future libraries may want to use other criteria. The authors then still use this insensible 0.01 TPM threshold in Figure 3a, which is not doing a universal analysis but rather looks per-cell-line. In the rebuttal, they seem to agree that TPM of 0.1 is a much more reasonable threshold; I don't understand why they still use 0.01 per cell line in Fig. 3a. The justification seems to again come from the analysis that looks at the `_median_` across all the CCLE, which doesn't make sense. Fig. 3a should show TPM >0.1 or TPM >1 results, not TPM >0.01 , which doesn't make any sense in a single cell line.

We would like to clarify again that this threshold was never used for the selection of lncRNAs included in library. Thus, there is no need redesign the library or clarify retrospectively this point.

Regarding the reviewer's comment on the use of median values: we previously used the median values for visualization purposes (easy accessibility). We do agree, however, that the really critical criterion is the TPM value for individual cell lines. We have therefore changed the relevant figure (Supplementary Fig. 3b), which is now displaying individual curves for all cell lines. Importantly, the values for individual cell lines behave like the median.

Because the median TPMs behave like the TPM of individual cell lines, displaying data for TPM >0.01 in the main figure (figure 3a) is still justified. We have therefore maintained figure 3a. TPM >0.01 distinguishes between non-expressed and expressed lncRNAs in individual cell lines (even though this criterion to define the "expressed" group includes lncRNAs with very low expression). Data for other TPM values (>0.1 , >1) are shown in Supplementary Fig. 3c and explained in detail in the text, as suggested by the reviewer. We think that this topic is now covered in extensive detail in the manuscript.

4. Lines 226-227, I don't understand what lncRNA complexity has to do with lncRNA secondary structure. One refers to the complexity of the splicing, and another to the folding of the RNA. Longer RNAs have more potential to have some structured regions, but the sentence in current form is confusing.

We agree. We changed the sentence accordingly: "Indeed longer lncRNAs have a higher propensity form structured/functional regions"

5. For the consistency between the effects of different gRNA arrays targeting the same lncRNA, the authors currently show in Fig. 6c the % of gRNAs that contribute to the phenotype, which is a confusing number (I don't quite understand what this means). A more intuitive metric would be to compute per-array log₂fc and then the authors can compare these between gRNA arrays targeting the same gene vs. those targeting different genes.

We agree. We changed the figure as suggested. Supplementary Fig. 6c.

6. Line 300: “saturation mutagenesis” is a term used to refer to mutations of all possible bases in a sequence. Here, this is not what the authors are doing (in fact, they are not doing any mutagenesis of all), therefore, “more exhaustive/comprehensive targeting” would be more appropriate here.

As suggested, we removed the term “saturation”.

7. Figure 5c – the terminology of “off-targets” is confusing, as the authors don’t know that these are off-targets, these can be merely suspected to be off-targets because they are close to other genes. “% close to other genes” is more accurate

As suggested, we now use the term “potential off-targets”.

8. Fig. 5f-g – statistical significance should be reported.

Now we included the statistical significance.

Reviewer #2 (Remarks to the Author):

The authors have adequately addressed most of my previous concerns and the work looks much more consolidated than the previous submission. This study provides a very useful platform/resource to screen functional lncRNAs in various characteristics of cancer. The following comments may help to further clarify the manuscript.

1. For all supplementary tables, legends should be provided to clarify the contents; e.g. in Supplementary Table 4, I'm not sure if everyone understands what these transcript IDs correspond to? Are these ID from ENSEMBL or what? in Supplementary Table 1, these contents make no sense without proper format and legends.

This information was available in the submitted “read me” document. We will of course again provide this information when resubmitting the manuscript.

Regarding the reviewer’s specific questions: IDs are not from ENSEMBL, because our collection includes lncRNAs derived from many other databases. We defined lncRNA genes by using our custom annotation. For each of these genes we provide the IDs of related transcripts as available in RNA central. For selected genes (our screening hits) we also provide the gene symbols from geneCard. We have added this clarification to the “read me” document.

2. The authors claimed in Line 199 that “the 5' RNA cap or the poly-A tail - structures that are essential for maintaining RNA stability”. There's no evidence to support this. Can RNA-seq genome browser track or some examples be shown to reveal the degradation of some targeted lncRNAs?

The RNA-seq approach used for these experiments (3' RNA-Seq, not full-length RNA-Seq) do not support the analyses proposed by the reviewer. However, the importance of 5' RNA cap or poly-A tail structures for maintaining RNA stability is well established in the field. We now added relevant evidence/references (Passmore, L. A. and Collier, J., Nat. Rev. Mol. Cell Biol., 2022. Collier, J. and Parker, R., Annu. Rev. Biochem., 2004. Decher, C. J. and Parker, R., Trends Biochem. Sci., 1994).

3. To respond to my previous point 6, the authors claimed that 24.4% (50 lncRNAs) were validated by orthogonal methods. Apparently, they meant by other studies. However, there's no citation or list of these 50 lncRNAs. Where is this number from?

The reviewer might have overlooked this. This information (list of 50 lncRNAs, relevant citations) was provided in the Supplementary Table 13.

4. Will the authors consider sharing these resources in public reservoir (e.g. Addgene) to benefit more users?

All material generated will be publicly available after publication. For each reagent, we have already created a pre-deposition number at Addgene.

Reviewer #3 (Remarks to the Author):

In the revised version of the manuscript, the authors substantially improved manuscript readability and sufficiently addressed my concerns and comments. I believe the manuscript is now ready to be published

This email has been sent through the Springer Nature Tracking System NY-610A-NPG&MTS

Final Decision Letter:

Dear Roland,

I am pleased to inform you that your Article, "Genome-scale pan-cancer interrogation of lncRNA dependencies using CasRx", has now been accepted for publication in Nature Methods. The received and accepted dates will be 6th Mar 2023 and 19th Jan 2024. This note is intended to let you know what to expect from us over the next month or so, and to let you know where to address any further questions.

Over the next few weeks, your paper will be copyedited to ensure that it conforms to Nature Methods style. Once your paper is typeset, you will receive an email with a link to choose the appropriate publishing options for your paper and our Author Services team will be in touch regarding any additional information that may be required. It is extremely important that you let us know now whether you will be difficult to contact over the next month. If this is the case, we ask that you send us the contact information (email, phone and fax) of someone who will be able to check the proofs and deal with any last-minute problems.

Please note that *Nature Methods* is a Transformative Journal (TJ). Authors may publish their

research with us through the traditional subscription access route or make their paper immediately open access through payment of an article-processing charge (APC). Authors will not be required to make a final decision about access to their article until it has been accepted. [Find out more about Transformative Journals](https://www.springernature.com/gp/open-research/transformative-journals)

Authors may need to take specific actions to achieve [compliance with funder and institutional open access mandates](https://www.springernature.com/gp/open-research/funding/policy-compliance-faqs). If your research is supported by a funder that requires immediate open access (e.g. according to [Plan S principles](https://www.springernature.com/gp/open-research/plan-s-compliance)) then you should select the gold OA route, and we will direct you to the compliant route where possible. For authors selecting the subscription publication route, the journal's standard licensing terms will need to be accepted, including [self-archiving policies](https://www.springernature.com/gp/open-research/policies/journal-policies). Those licensing terms will supersede any other terms that the author or any third party may assert apply to any version of the manuscript.

If you are active on Twitter/X, please e-mail me your and your coauthors' handles so that we may tag you when the paper is published.

Please note that you and any of your coauthors will be able to order reprints and single copies of the issue containing your article through Nature Portfolio's reprint website, which is located at

<http://www.nature.com/reprints/author-reprints.html>. If there are any questions about reprints please send an email to author-reprints@nature.com and someone will assist you.

Best regards,
Madhura

Madhura Mukhopadhyay, PhD
Senior Editor
Nature Methods